# Inference-Time Hyper-Scaling
# with KV Cache Compression

**Adrian Łańcucki**[†]    **Konrad Staniszewski**[†○]    **Piotr Nawrot**[◇*]    **Edoardo M. Ponti**[†◇]

[†]NVIDIA    [○]University of Warsaw    [◇]University of Edinburgh

## Abstract

Inference-time scaling trades efficiency for increased reasoning accuracy by generating longer or more parallel sequences. However, in Transformer LLMs, generation cost is bottlenecked by the size of the key–value (KV) cache, rather than the number of generated tokens. Hence, we explore inference-time *hyper-scaling*: by compressing the KV cache, we can generate more tokens within the same compute budget and further improve the accuracy of scaled inference. The success of this approach, however, hinges on the ability of compression methods to preserve accuracy even at high compression ratios. To make hyper-scaling practical, we introduce Dynamic Memory Sparsification (DMS), a novel method for sparsifying KV caches that only requires 1K training steps to achieve $8\times$ compression, while maintaining better accuracy than training-free sparse attention. Instead of prematurely discarding cached tokens, DMS delays token eviction, implicitly merging representations and preserving critical information. We demonstrate the effectiveness of inference-time hyper-scaling with DMS on multiple families of LLMs, showing that it boosts accuracy for comparable inference latency and memory load. For instance, we enhance Qwen-R1 32B by 12.0 points on AIME 24, 8.6 on GPQA, and 9.7 on LiveCodeBench on average for an equivalent number of memory reads.

## 1   Introduction

Scaling inference-time compute—employed in models such as OpenAI's o1 (OpenAI et al., 2024) or DeepSeek's R1 (Guo et al., 2025)—trades off increased inference time and memory for higher reasoning accuracy in large language models (LLMs). Models reason by generating intermediate steps that explore the problem before reaching an answer. Adjusting the depth and breadth of this exploration—known as sequential and parallel scaling, respectively (Muennighoff et al., 2025)—controls the inference-time compute budget (Yao et al., 2023; Uesato et al., 2022; Wang et al., 2023; Lightman et al., 2024).

Despite its success, scaling inference-time compute is fundamentally bottlenecked in Transformer LLMs by the number of tokens from the key–value (or KV) cache that are attended to during auto-regressive generation. This cache grows linearly with respect to the length and number of reasoning chains, as the new key–value representations are appended to it.

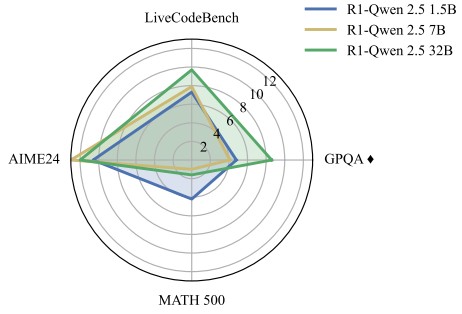

Figure 1: Average absolute gains of DMS over the original LLMs during inference-time scaling on reasoning tasks for the same KV cache memory reads, (a proxy for latency).

---

[*]Work done as an intern at NVIDIA.

Hence, it can easily exhaust the memory of the accelerator and slow down each generation step, as attention is memory-bound: its cost is dominated by the time needed to retrieve the cache from memory. Fortunately, several methods can mitigate these issues during post-training or inference. These rely on training-free heuristics to sparsify the KV cache (Oren et al., 2024; Li et al., 2024), selectively retrieve subsets of tokens (Tang et al., 2024), or retrofit LLMs with the ability to choose whether to merge or append items to the cache (Nawrot et al., 2024).

In this work, we investigate for the first time whether efficient attention methods enhance inference-time scaling. In principle, by exploring more concurrent reasoning threads or longer threads for the same memory or runtime budget, efficient models can achieve higher-quality predictions than their original counterparts. However, this hinges upon the crucial assumption that efficient attention does not degrade the model's reasoning abilities, which unfortunately is often the side effect of training-free sparsification methods (Zhang et al., 2023a; Oren et al., 2024). On the other hand, KV cache compression during post-training usually better preserves the model's quality, but also demands costly retrofitting procedures (Nawrot et al., 2024).

In order to overcome these limitations, as a second main contribution, we propose Dynamic Memory Sparsification (DMS), a new method that combines the best of both worlds by retrofitting LLMs to sparsify the KV cache through an inexpensive procedure. We thus demonstrate that sparsification—rather than more complex token merging proposed in Dynamic Memory Compression (DMC; Nawrot et al., 2024)—is sufficient to retain accuracy at high compression ratios. This, in turn, allows us to retrofit LLMs with KV cache compression in a much more sample-efficient way than DMC, achieving $8\times$ compression with only 1K training steps. On the other hand, the superior performance of DMS highlights the benefits of retrofitting LLMs over training-free heuristics.

We evaluate inference-time scaling capabilities of efficient attention (including DMS) on reasoning datasets such as MATH-500 (Hendrycks et al., 2021b) and AIME 2024 for math, GPQA Diamond (Rein et al., 2024) for hard sciences, and LiveCodeBench (Jain et al., 2025) for coding. We find that DMS significantly improves the Pareto frontiers across various model sizes and datasets, outperforming vanilla LLMs in both memory reads (which is a proxy for runtime) and peak memory use. Notably, DMS consistently dominates other baselines for efficient attention, which we also verify on a broader set of tasks outside of inference-time scaling. DMS variants even surpass the corresponding vanilla LLMs on long-context tasks, such as needle-in-a-haystack and variable state tracking (Hsieh et al., 2024), while achieving higher efficiency. Overall, this validates the effectiveness of efficient attention—unlocked by DMS—for inference-time scaling, which improves the reasoning capabilities of models under any given inference-time budget.

## 2 Background

### 2.1 Inference-time Scaling

Inference-time scaling allows a model to 'think longer or more broadly' about a problem to enhance the quality of its prediction, by leveraging extra compute during generation (Du et al., 2024; Madaan et al., 2023; Yao et al., 2023). In practice, when presented with a prompt $\mathbf{x}$, a Large Language Model $f_{\text{LLM}}$ can explore $n$ chains of reasoning $[\mathbf{z}_1, \ldots, \mathbf{z}_n]$ to generate corresponding answers $[\mathbf{y}_1, \ldots, \mathbf{y}_n]$. While some strategies involve guiding this exploration through a Process Reward Model (PRM; Li et al., 2023; Feng et al., 2023; Lightman et al., 2024; Uesato et al., 2022; Wang et al., 2024a; Snell et al., 2024) by scoring each reasoning step, recent systematic comparisons established that simpler PRM-free strategies such as majority voting (Wang et al., 2025b) remain the most competitive.

Hence, scaling can be easily achieved in two ways: increasing the maximum length for chains of reasoning (known as *sequential* scaling) or increasing their number (known as *parallel* scaling). These two quantities can be controlled to set a 'token budget' for inference-time computation (Muennighoff et al., 2025), which determines the memory load and latency. In fact, in Transformer LLMs, the KV cache grows linearly with the number of generated tokens. Crucially, it is stored on VRAM in GPU accelerators, contributing significantly to the overall memory load. At the same time, the larger the token budget, the more retrieving the KV cache through high-bandwidth memory access dominates latency during generation. As a consequence, the KV cache constitutes a bottleneck for inference-time scaling. This leads to the natural question: by making the KV cache leaner, could we scale the length and number of reasoning threads and enhance the accuracy of existing LLMs for an equivalent compute budget?

## 2.2 Training-free KV Cache Eviction

An intuitive strategy to reduce the size of the KV cache is to evict tokens, i.e., dynamically remove the key–value pairs of the least relevant tokens during inference. Recent methods have addressed this challenge by selectively managing tokens within a sliding window of context of size $w$. For instance, for each time step $t$, TOVA (Oren et al., 2024) evicts the token with the lowest attention weight such that $i_{\text{TOVA}} = \min_i \sum_{h \in H} a_h(t)_i$ where $a_h(t)_i$ denotes the attention weight assigned to token $i$ by attention head $h$ at time step $t$. Similarly, Heavy-Hitter Oracle (H2O; Zhang et al., 2023a) evicts the token with the lowest *cumulative* attention, additionally keeping a sliding window of recent tokens. This family of approaches (more are surveyed in Appendix B) incur minimal computational overhead due to their efficient heuristics for eviction scores, while retaining a maximum KV cache size of $w$.

A different strategy is adopted by Quest (Tang et al., 2024), which fully retains the entirety of the KV cache but only retrieves the most relevant *pages* (i.e., fixed-size blocks of contiguous KV items) from memory. Relevant pages are determined through a heuristic that approximates attention scores from the highest-magnitude dimensions of each page's KV items. While this approach accelerates generation by reducing memory transfers without permanently evicting tokens, it does not reduce memory load. In fact, to efficiently perform page selection, the method requires storing additional page representations, resulting in a slight memory overhead rather than savings.

## 2.3 Learned KV Cache Compression

While mitigating latency and/or memory load, training-free KV cache eviction methods often incur performance degradation at high eviction rates. To overcome this limitation, Dynamic Memory Compression (DMC; Nawrot et al., 2024) reduces KV cache size by dynamically compressing representations, potentially extracting and retaining vital information. At each timestep $t$, for every attention head separately, DMC models decide to either *append* the new key–value pair to KV cache as in standard Transformers, or *merge* it into the most recent cache entry using weighted averaging. As a consequence, every attention head produces a uniquely compressed KV sequence with a possibly distinct compression ratio (CR). This flexibility allows the model to preserve critical information while aggressively compressing redundant representations, unlike KV cache eviction and sparse attention methods that mostly impose uniform compression budgets (Nawrot et al., 2025).

Applying DMC requires continued training of existing LLMs (a.k.a. 'retrofitting'), during which discrete decisions (append versus merge) are relaxed into continuous variables via stochastic reparameterization (Louizos et al., 2018), enabling gradient-based optimization. Although it requires a fraction of the pre-training budget, the computational cost is still significant; however, this helps retaining the original LLM quality as the model is trained to operate with a compressed KV cache.

# 3 Dynamic Memory Sparsification

Token eviction strategies effectively reduce KV cache size, but degrade downstream performance at higher eviction rates. Conversely, DMC offers robust compression at the cost of expensive continued training. To scale inference-time efficiency even further, it is therefore essential to develop a KV cache compression method that is inexpensive, easy to integrate, and maintains accuracy at high compression ratios. To this end, we propose Dynamic Memory Sparsification (DMS), a method of teaching pre-trained models a simple, adaptive token eviction policy. As such, it combines the advantages of eviction and trained compression, with significantly higher data efficiency than DMC.

## 3.1 Retrofitting Models with DMS

To develop DMS, we follow Nawrot et al. (2024)'s pipeline to retrofit pre-trained LLMs rather than training them from scratch. We introduce two crucial modifications: (i) whereas DMC merges (weighted-averages) tokens, DMS simply evicts them; and (ii) we separate the time of eviction decisions from the time of their execution–when a token is flagged for eviction, the model is given a number of generation steps to integrate it's information. Below, we describe the procedure for a single attention head, though the same process is applied across all KV heads independently.

**Eviction Decisions** Given an input hidden vector to an attention layer $\mathbf{h}_t$ at inference time step $t$, DMS predicts a binary eviction decision $\alpha_t$ which controls the eviction of the key–value pair $(\mathbf{k}_t, \mathbf{v}_t)$.

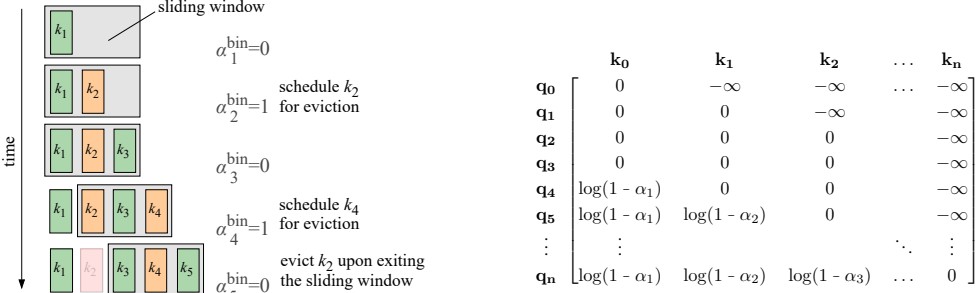

|  | $\mathbf{k_0}$ | $\mathbf{k_1}$ | $\mathbf{k_2}$ | $\dots$ | $\mathbf{k_n}$ |
|---|---|---|---|---|---|
| $\mathbf{q_0}$ | $0$ | $-\infty$ | $-\infty$ | $\dots$ | $-\infty$ |
| $\mathbf{q_1}$ | $0$ | $0$ | $-\infty$ | | $-\infty$ |
| $\mathbf{q_2}$ | $0$ | $0$ | $0$ | | $-\infty$ |
| $\mathbf{q_3}$ | $0$ | $0$ | $0$ | | $-\infty$ |
| $\mathbf{q_4}$ | $\log(1 - \alpha_1)$ | $0$ | $0$ | | $-\infty$ |
| $\mathbf{q_5}$ | $\log(1 - \alpha_1)$ | $\log(1 - \alpha_2)$ | $0$ | | $-\infty$ |
| $\vdots$ | $\vdots$ | | | $\ddots$ | $\vdots$ |
| $\mathbf{q_n}$ | $\log(1 - \alpha_1)$ | $\log(1 - \alpha_2)$ | $\log(1 - \alpha_3)$ | $\dots$ | $0$ |

(a) DMS key cache management during inference.      (b) Attention mask $M_\alpha$ during training.

Figure 2: During each inference step (**left**) the incoming key–value pair $(\mathbf{k}_t, \mathbf{v}_t)$ might be selected for later eviction, based on predicted binary decisions $\alpha^{\mathrm{bin}} \in \{0, 1\}$ (we show only a sequence of keys for clarity). The eviction takes place as soon as the pair falls out of the sliding window. During training (**right**), this behavior is induced with an additive attention mask. Eviction decisions are relaxed from binary to continuous $\alpha \in [0, 1]$.

To maintain differentiability during training, $\alpha_t$ is learned through stochastic reparametrization with a Gumbel-sigmoid distribution as a gradient estimator:

$$\alpha_t \sim \text{Gumbel-sigmoid}(\mathbf{h}_t \mathbf{w}^\top + b, \tau) \quad \alpha_t \in [0, 1], \tag{1}$$

where $\mathbf{w} \in \mathbb{R}^d$ is a vector of trainable weights initialized as $\mathbf{w} = [0, \dots, 0]^\top$. In addition, we set a low temperature $\tau$ to encourage discrete eviction decisions and $b = -5$ in order to offset the logits and initiate training with $\alpha_t \approx 0$, preventing eviction early in training. Empirically, this configuration prevents initial loss spikes, which might cause catastrophic forgetting (Nawrot et al., 2024).

During training, a sequence of eviction decisions $\alpha_{1:T}$ is used to construct a mask $M_\alpha \in (-\infty, 0]^{T \times T}$ (Figure 2b), which is added to unnormalized attention scores $QK^\top$. The elements that are not part of the causal mask (set to $-\infty$) are set to $\log(1 - \alpha_t)$. The mask selectively modulates token visibility: $\alpha_t = 1$ fully masks a token, $\alpha_t = 0$ indicates no masking, and values in between make a token only partly accessible. It follows that evicting a particular $\mathbf{k}_i$ entails evicting the corresponding $\mathbf{v}_i$.

**Delayed Eviction via Sliding Window**   Immediate eviction can harm the model's abilities by prematurely discarding useful context. To mitigate this, we propose delaying the execution of eviction decisions. Specifically, the eviction decision $\alpha_t$ is made at timestep $t$, but the token selected for eviction remains available until a future timestep $t + w$. This delay creates a sliding window of size $w$ and is implemented by setting positions within the window to 0 when constructing $M_\alpha$.

Previous work indicates that decoder-only models heavily attend to recent tokens (Xiao et al., 2024; Jiang et al., 2024). Consequently, delayed eviction enables the model to extract relevant information from such tokens before their removal. Foreshadowing Section 5.3, we find that immediate eviction leads to rapid accuracy degradation , whereas delayed eviction maintains stable training, dramatically reducing the number of training tokens needed to achieve a given CR.

**Training Objective**   During training we follow DMC and apply a one-sided $\ell_1$ loss term which forces the model to match the average value of predicted $\alpha$ for a given input to the target compression $\alpha^\star$, i.e., $\mathcal{L}_{\text{aux}} = \max\left(\alpha^\star L H T - \sum_{l \in L} \sum_{h \in H} \sum_{t \in T} \alpha_{lht}, 0\right)$, where $L, H, T$ denote the number of layers, KV attention heads, and sequence length, respectively. Over the course of training, the target compression $\alpha^\star$ is linearly annealed from 0 to $\left(1 - \frac{1}{\text{CR}}\right)$. We train the model using a logit distillation loss $\mathcal{L}_{\text{D}}$ loss (Hinton et al., 2015), described in detail in Section 4. The distillation loss and auxiliary loss are then combined into a single objective: $\mathcal{L} = \mathcal{L}_{\text{D}} + \mathcal{L}_{\text{aux}}$. Since we do not enforce any constraints on compression for individual attention heads, they adopt possibly different compression ratios and produce KV sequences of possibly different lengths.

**Performance Considerations**   The overhead of DMS on the attention mechanism comes solely from constructing and applying the additive attention mask, which never needs to be materialized.

For each attention head, it can be compactly passed as a vector of eviction decisions $\alpha_{1:T}$, and is implementable with existing tools (Wang et al., 2025a; Dong et al., 2024). Implementation-wise, a neuron is re-purposed from $\mathbf{q}_t$ or $\mathbf{k}_t$ to predict $\alpha_t$ instead of adding a parameter vector $\mathbf{w}$ for every attention head (Nawrot et al., 2024). Hence, no extra parameters are added.

## 3.2 Inference

Figure 2a shows the inference time operation of DMS. The decision variables are rounded to the nearest integer $\alpha_t^{\mathrm{bin}} = \lfloor \mathrm{sigmoid}(\mathbf{h}_t \mathbf{w}^\top + b) \rceil \in \{0, 1\}$. If $\alpha_t^{\mathrm{bin}} = 1$, then the $(\mathbf{k}_t, \mathbf{v}_t)$ pair needs to be evicted at time $t + w$. The sparsity introduced by DMS is also leveraged during the prefilling phase to eliminate unnecessary computation (Wang et al., 2025a).

Performance-wise, DMS does not introduce any new read/write operations on the KV cache, since the evicted tokens could be simply overwritten by incoming ones, under the assumption that the keys are stored in the KV cache with positional information. PagedAttention (Kwon et al., 2023) facilitates storing the sparsified KV cache in memory, where pages are allocated to individual attention heads. This formulation enables our reuse of existing, efficient kernels that support PagedAttention.

## 4 Experimental Setup

**Models and Baselines**   To evaluate inference-time scaling through KV cache compression, we primarily focus on reasoning models of different sizes, including Qwen 2.5 1.5B, 7B, and 32B distilled from DeepSeek R1 (Guo et al., 2025) and Qwen3-8B distilled from Qwen3-235B-A22B (Yang et al., 2025). In addition, as a sanity check on other families of models and for initial ablations on method design, we test the accuracy of efficient attention methods also on Llama 3.2 1B Instruct (Grattafiori et al., 2024). We retrofit all these models with DMS and compare them against the original models, DMC, and training-free KV cache sparsification methods described in Section 2.2: Token Omission via Attention (TOVA; Oren et al., 2024), Heavy-Hitter Oracle (H2O; Zhang et al., 2023a), and Quest (Tang et al., 2024).[2] Crucially, all the LLMs included in the experiments use Grouped Query Attention (GQA; Ainslie et al., 2023), hence KV tokens are shared among multiple query heads. This exacerbates the destructive effects of training-free token eviction.

**Logit Distillation and Retrofitting**   In contrast to conventional retrofitting methods employing standard language modeling loss (Nawrot et al., 2024), we retrofit all models through logit distillation (Hinton et al., 2015). In particular, the original LLM acts as the teacher and the DMS-retrofitted one as the student. As previously observed in other settings (Sreenivas et al., 2024; Minixhofer et al., 2025), we found that logit distillation provides greater robustness to shifts in training data, since the original data mixtures are rarely public, and is especially beneficial for fragile LLMs with lower parameter counts. We provide information on the training data for distillation in Appendix E.

The retrofitting process is inspired by DMC (Nawrot et al., 2024). The amount of required data depends directly on the context length of retrofitted models and the target compression ratio: higher ratios necessitate larger datasets. We employ a linear schedule that runs for 100 training steps for each unit of compression ratio: $\mathrm{CR}(t) = \frac{t}{100} + 1$. Crucially, annealing the CR generates a family of models with different compression ratios from a single retrofitting run. Unless otherwise stated, we train DMS models with a sliding window—and equivalently an eviction delay—of 256 tokens.

## 5 Results

### 5.1 Inference Time Hyper-Scaling

**Goal and Metrics**   We aim to determine whether KV cache compression increases downstream performance by effectively leveraging a larger 'token budget' for equivalent latency and memory load compared with vanilla Transformers. We run our experiments for inference-time hyper-scaling on datasets that require advanced reasoning capabilities, following Snell et al. (2024) and Guo et al. (2025). Specifically, we evaluate AIME 24 and MATH-500 (Hendrycks et al., 2021a) for math problems, GPQA Diamond (Rein et al., 2024) for physics, chemistry, and biology, and LiveCodeBench

---

[2]Our baseline implementations closely follow the publicly available reference implementations.

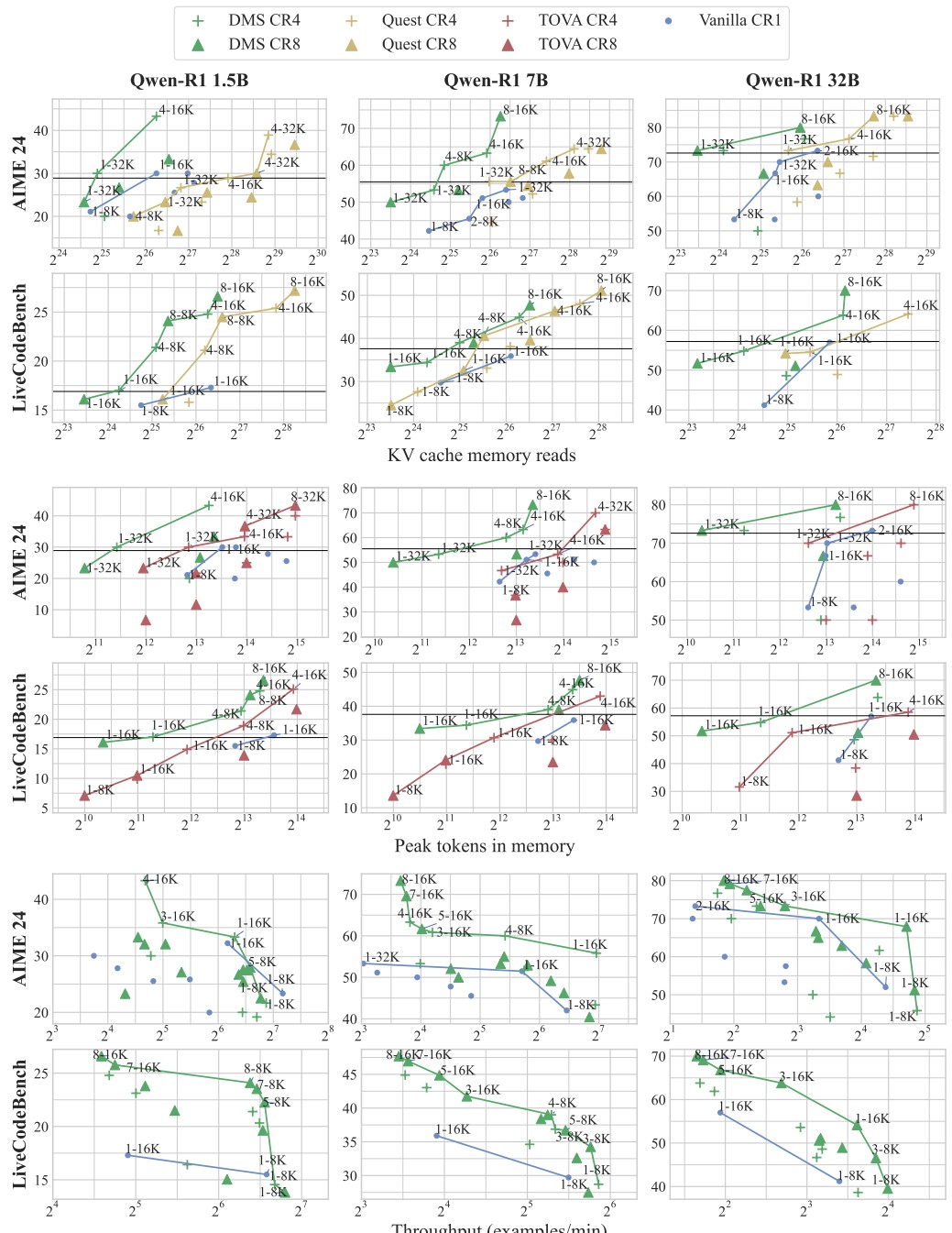

Figure 3: **Inference-time scaling results** comparing exact-match accuracy ($y$-axis) against performance metrics ($x$-axis). Point colors indicate the compression algorithm used, shapes the compression ratio, and W–L labels denote the scaling strategy (W: number of sampled reasoning threads; L: sequence length). Colored lines indicate the respective Pareto frontiers. The horizontal black lines mark the accuracy reported by Guo et al. (2025) for the 1–32K vanilla model. **Top:** A comparison in terms of KV-cache token reads, used as an implementation-agnostic proxy for attention compute. **Middle:** A comparison in terms of the peak number of tokens in memory, reflecting memory load. **Bottom:** Throughput calculated at the maximum batch size that accommodates the corresponding W–L configuration. Across plots, DMS attains the best Pareto frontiers, indicating that KV-cache compression is an effective strategy for improving inference-time scaling.

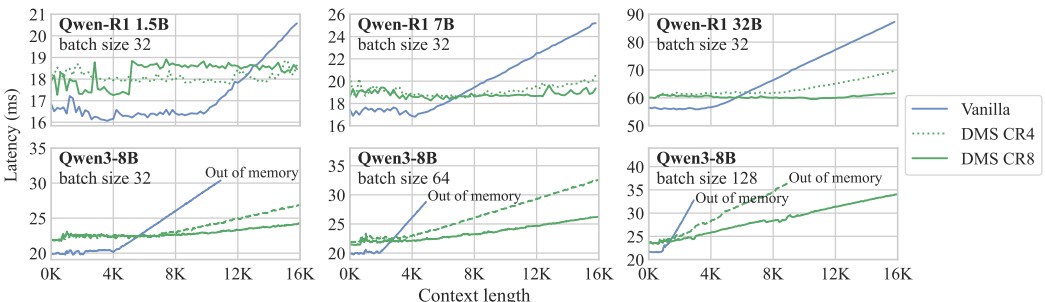

Figure 4: **Latency of models** ($y$-axis) at different context lengths ($x$-axis). **Top:** We compare the effect of different model sizes (Qwen-R1 1.5B, 7B, 32B) for the same batch size (32). **Bottom:** We compare the effect of different batch sizes (32, 64, 128) for the same model (Qwen3-8B). Batch size reflects both the number of parallel reasoning threads and the number of queries the model is serving. These plots show that inference becomes memory-bound at different context lengths depending on model scale and batch size, yielding distinct accuracy–efficiency trade-offs.

(Jain et al., 2025) for coding. As the performance metric, we use exact match after mapping outputs to a unified math representation (for MATH-500 and AIME 24) or to one of the four available choices (for GPQA Diamond). For LiveCodeBench, we report pass@all, i.e., we count a success if any generated sequence passes the tests.

As metrics for the effective budget in time and memory, we focus first on two implementation-agnostic quantities: (i) **KV-cache token reads**, the total number of items in the KV cache attended to at each generation step, summed across steps. This reflects runtime efficiency, as loading the KV cache from memory quickly becomes a main bottleneck during generation, contributing a share of inference latency that increases with sequence length (Tang et al., 2024; Nawrot et al., 2025). Second, (ii) **peak tokens in memory**, which represents the maximum KV-cache size, critical for memory-constrained hardware such as GPUs or edge devices. Finally, (iii) **throughput** at the maximum batch size that accommodates each tested configuration.

To establish a range of trade-offs between downstream performance and compute, we run experiments with varying budget configurations in terms of maximum the number of parallel reasoning chains or *width* (W), sequence *length* (L), and *compression ratio* (CR), defining each setting as a tuple W-L-CR, where CR $1\times$ denotes vanilla models. By identifying the Pareto frontier for each method, we determine which ones offer superior performance for the same budget. We report results for AIME 24 and LiveCodeBench in Figure 3, and additional results on MATH-500 and GPQA Diamond in Appendix D. For simplicity, as baselines we report only the state-of-the-art training-free methods: Quest for accuracy-–latency and TOVA for accuracy—memory load.

**Accuracy vs. Memory Reads and Peak Memory**    From Figure 3, and additionally Appendix D and Figure 7, we observe that KV cache compression methods generally yield superior Pareto frontiers compared to vanilla LLMs across model sizes and datasets. Specifically, the best-performing method in each dataset–size combination substantially improves the scores at comparable memory transfer budgets (which drive latency). Averaging Pareto frontier margins across budgets, as detailed in Appendix G and Table 8 and summarized in Figure 1, we find average gains for DMS of 12.5 for AIME 24, 2.3 for MATH-500, 5.8 for GPQA Diamond, and 8.3 for LiveCodeBench. Variability across datasets primarily reflects their saturation levels; for instance, models achieve very high performance on MATH-500 even with limited budgets. Notably, performance gains from DMS decrease with increasing model scale on MATH-500, yet increase with scale on GPQA Diamond and LiveCodeBench. Overall, these findings indicate that KV cache compression exhibits more favorable scaling properties than full KV retention in vanilla LLMs, highlighting its potential for advancing their reasoning capabilities.

Moreover, comparing DMS with other KV cache compression methods, we find that its Pareto frontier clearly dominates the best baselines for both efficiency metrics: Quest for KV cache memory reads and TOVA for peak tokens in memory (Figure 3). This is even more remarkable considering that Quest sacrifices memory efficiency by fully preserving the KV cache to mitigate accuracy

Table 1: Evaluation of Llama 3.2 1B Instruct on a broader array of tasks, across different methods and compression ratios (CR). We note that due to full-dense attention prefill, Quest is equivalent to vanilla on MMLU and HellaSwag. The DMS model used in this comparison was trained with a sliding window of just 16 tokens. As datasets, we include GSM8K (Cobbe et al., 2021) for grade-school math, MMLU (Hendrycks et al., 2021a) for factuality, HellaSwag (Zellers et al., 2019) for zero-shot common-sense question answering, and Needle in a Haystack (NIAH; Kamradt, 2023) and Variable Tracking (VT; Hsieh et al., 2024) for long context processing.

| CR | 1× | 2× | | | | | 3× | | | | | 4× | | | | |
|---|---|---|---|---|---|---|---|---|---|---|---|---|---|---|---|---|
| Method | Vanilla | H2O | TOVA | Quest | DMC | DMS$^{win=16}$ | H2O | TOVA | Quest | DMC | DMS$^{win=16}$ | H2O | TOVA | Quest | DMC | DMS$^{win=16}$ |
| **GSM8K** | 47.0 | 44.0 | 45.0 | 45.1 | 31.9 | **46.9** | 32.9 | 40.1 | 44.7 | 6.4 | **46.5** | 14.7 | 20.2 | 39.9 | 3.6 | **42.3** |
| **MMLU** | 47.9 | 45.7 | 43.4 | 47.9 | 34.9 | **48.0** | 37.6 | 38.1 | **47.9** | 26.3 | 45.2 | 32.7 | 35.2 | **47.9** | 25.6 | 40.3 |
| **HellaS** | 43.4 | 42.9 | 42.8 | **43.4** | 42.2 | 43.3 | 42.1 | 42.5 | **43.4** | 40.0 | 43.3 | 41.3 | 41.8 | **43.4** | 39.4 | **43.4** |
| **NIAH** | 96.4 | 34.0 | 65.2 | 95.8 | 0.0 | **97.8** | 17.2 | 40.2 | **95.6** | 1.8 | 93.6 | 13.4 | 28.0 | 95.8 | 0.0 | **96.8** |
| **VT** | 55.8 | 27.4 | 56.2 | 53.0 | 0.0 | **63.2** | 17.6 | 45.2 | 50.4 | 0.2 | **69.2** | 12.6 | 33.8 | 49.6 | 4.0 | **67.6** |

degradation—and yet DMS still offers a better latency–accuracy trade-off. Datasets like MATH-500, where Quest's Pareto frontier mostly overlaps with vanilla at all scales, illustrate that gains from larger token budgets can be eaten away, unless performance is retained even at high CRs. DMS meets this desideratum in a data-efficient way, thus offering inexpensive hyper-scaling with existing LLMs.

Zooming in on specific results, we can assess which W-L-CR configurations tend to lie on the Pareto frontier for DMS. For most tasks, these consist of a combination of sequential and parallel scaling, hinting at the necessity of using both for inference-time scaling. Moreover, most DMS points (for CRs of 4× and 8×) lie on the Pareto frontier, indicating that even higher compression retains sufficient quality to afford superior trade-offs.

**Latency and Throughput Measurements** Next, we investigate how the reduced memory reads and peak memory of DMS translate into latency and throughput, measured on an NVIDIA H100 SXM GPU in 16-bit precision, using a simple implementation based on the Hugging Face Transformers library and FlashAttention (Dao, 2024). First, we illustrate the latency of a single generation step of DMS and vanilla Qwen-R1 models for different context lengths in Figure 4. Initially, latency is roughly constant, but it rises early due to the increasing cost of reading the KV cache. The exact context length at which this occurs depends primarily on token batch size, which reflects both the number of queries served and the number of reasoning traces per query in parallel scaling. In addition, the vanilla LLM can exhaust VRAM quickly at high batch sizes. For more details, consult Appendix I.

In real-world scenarios, LLM systems should serve as many queries as possible while retaining high quality. Hence, we compare the throughput of DMS and vanilla LLMs. As Figure 3 (bottom) illustrates, at equivalent accuracy on AIME 24 and LiveCodeBench (determined by the specific inference-time hyper-scaling configuration), DMS models can serve significantly more queries in parallel when using the maximum batch size that fits in memory. This demonstrates that the gains observed in Figure 3 translate into effective speedups when deploying LLM systems with DMS.

### 5.2 DMS for General-purpose LLMs

Moreover, we aim to establish whether DMS is effective beyond inference-time scaling settings, so that a model retrofitted with DMS can be reliably deployed as a general-purpose LLM. To this end, we first compare DMS with respect to vanilla models for equivalent generated token lengths (rather than actual compute budget). We focus again on the same models and datasets as Section 5.1. From Tables 10 to 12 in Appendix G, it emerges that DMS mostly preserves the original accuracy at CR 4× and yields minimal degradations at CR 8×.

Table 2: Evaluation of DMS 8× Qwen3-8B with a sliding window of 512 tokens.

| Benchmark | Think | Vanilla | DMS 8×$^{win=512}$ |
|---|---|---|---|
| GPQA Diamond | ✓ | 58.8 | 57.6 |
| MMLU-Pro | ✓ | 74.2 | 73.5 |
| AIME 2024 | ✓ | 75.0 | 73.0 |
| MATH-500 | ✓ | 95.1 | 95.5 |
| HumanEval | ✓ | 87.8 | 89.6 |
| IFEval | ✓ | 90.3 | 88.8 |
| ArenaHard v0.1 | ✓ | 88.4 | 89.7 |

We also benchmark Qwen3-8B on a broader set of tasks in Table 2. The tasks include math and science (GPQA Diamond, AIME 2024, MATH-500), factuality (MMLU-Pro Wang et al., 2024b), coding

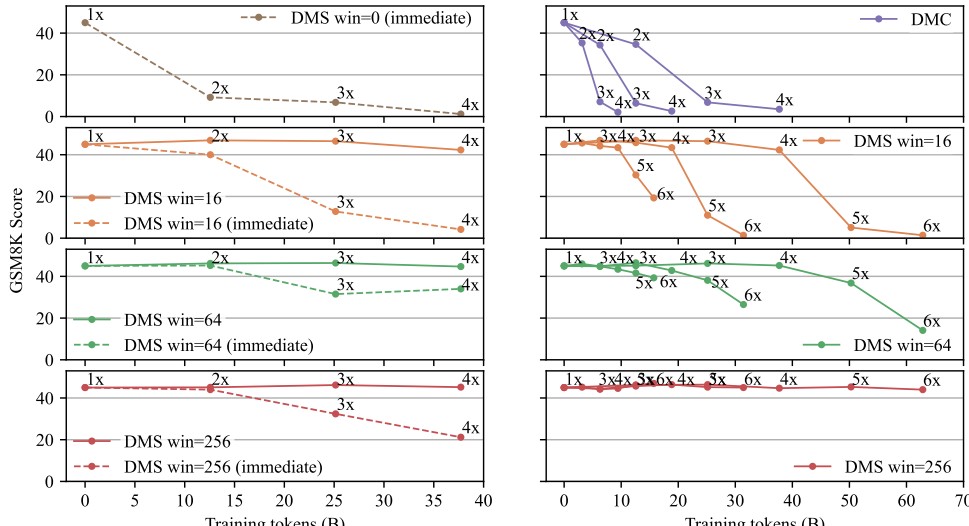

Figure 5: **GSM8K 0-shot scores** of Llama 3.2 1B Instruct across different compression variants. **Left:** delayed eviction (default) with a 16-token window consistently preserves reasoning abilities of the model, while immediate eviction causes rapid degradation. The quality gap only widens as the compression gets stronger. **Right:** DMS requires an order of magnitude less data to train than DMC. This was also observed for Qwen 2.5 R1 models with 1.5B, 7B, and 32B parameter scales.

(HumanEval; Chen et al., 2021), conversation (ArenaHard v0.1; Li et al., 2025), and instruction following (IFEval; Zhou et al., 2023). Technical details for the experimental setup are provided in Appendices G and H. From Table 2, it emerges that DMS is within close range of the accuracy of vanilla Qwen3-8B. Moreover, in Figure 8 we compare the vanilla and DMS Qwen3-8B models in terms of the accuracy–throughput trade-off during inference scaling on LiveCodeBench and find that DMS consistently matches the accuracy of vanilla, while allowing up to $5\times$ higher throughput.

Finally, as a way to ensure that DMS's sparse prefilling does not affect performance in short generation settings, we evaluate Llama 3.2 1B Instruct, a small, non-reasoning model, on a broad set of tasks (Table 1). In this setting DMS also stands out as the most robust method for accelerated inference. Overall, DMS's accuracy retention at high compression ratios makes it suitable not only for inference-time hyper-scaling but also for general-purpose use, independent of the context or generation length.

### 5.3 Ablations

The design choices in DMS were informed by results of small-scale experiments. We present ablations on eviction policy and data efficiency during retrofitting of the Llama 3.2 1B Instruct models. To evaluate the impact of *delayed* eviction, we trained additional models with *immediate* eviction, which aligns more closely with existing token eviction methods:

- **Delayed eviction:** determines the eviction of $(\mathbf{k}_t, \mathbf{v}_t)$ at a future time step $t + w$
- **Immediate eviction:** $\alpha_{t+w}$ determines the eviction of past $(\mathbf{k}_t, \mathbf{v}_t)$ at time step $t + w$

Both policies were tested with different sliding window sizes. Remarkably, DMS retains reasoning capabilities with a window of only 16 tokens up to a compression ratio of $4\times$, as shown in Figure 5. Larger sliding windows better preserve reasoning capabilities, which is expected in a zero-shot setting. In contrast, immediate eviction drastically deteriorates scores for every tested sliding window length.

Regarding data efficiency, the right panel of Figure 5 shows how scores vary when retrofitting with different training token budgets. Crucially, DMS achieves higher scores than DMC while using $8\times$ fewer training tokens. In practice, the reasoning models described in Section 5.1 were trained with $60\times$ less training data,[3] achieving CR $4\times$ within 300 training steps and CR $8\times$ within 700 steps.

---

[3]DMC was reported to require 44K training steps to reach CR8, with performance deteriorating when the amount of data is halved (Nawrot et al., 2024).

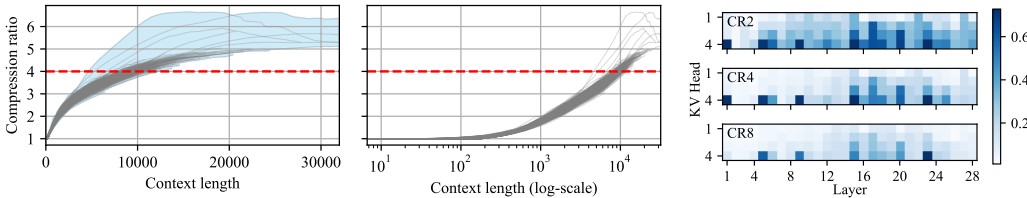

Figure 6: Left: **The measured compression ratio** for Qwen-R1 7B, trained with DMS CR $4\times$, while processing AIME 24, MATH-500, and GPQA Diamond problem instances. Right: **Average per-head compression** learned by the model, as a percentage of retained tokens sorted for every layer.

Finally, we measured how the CR varies for different lengths of the sequences generated through DMS (Figure 6 left). The resulting pattern closely matches that reported in Nawrot et al. (2024). The model sparsifies less than the target CR in early parts of a sequence, but even more aggressively than specified beyond 10K tokens. This behavior stems from the training objective and from the tendency of the conditional entropy rate of natural-language text to decrease as the context length grows. In Figure 6 (right), we also observe that early layers are compressed to a smaller degree than later layers.

## 5.4 Discussion

Research on inference-time scaling has so far mostly assumed an equivalence between compute budget and generated tokens, in terms of sequence length or parallel samples (Brown et al., 2024; Zhang et al., 2023b; Wang et al., 2023). This budget can be allocated adaptively to the complexity of the task (Snell et al., 2024) or forced to meet an amount pre-defined by the user (Muennighoff et al., 2025). To the best of our knowledge, our work is the first to fully disentangle generated tokens from the effective compute (latency and peak memory load) when reasoning in the discrete language space. In fact, we show how KV cache compression methods can effectively expand the token budget for the same compute budget. A separate family of strategies are based on latent space reasoning (Geiping et al., 2025), which add a recurrent block on top of Transformer LLMs; however, this effectively requires a separate architecture rather than boosting existing LLMs, and it remains unclear whether these scale similarly to reasoning in the discrete token space.

While in this work we opt for verifier-free scaling strategies, adopting the recommendations of Wang et al. (2025b), inference-time scaling can rely on process reward models (PRMs) to verify intermediate reasoning steps. This allows effective self-critique loops and re-ranking candidate solutions (Uesato et al., 2022; Lightman et al., 2024; Liang et al., 2024). Nonetheless, we remark that hyper-scaling can be extended to PRM strategies, too. In particular, the verifier's complexity is quadratic in the sequence length; to complement the benefits of KV cache compression of the LLM, the PRM would need to be accelerated by prefilling-time sparse attention methods, such as MInference (Jiang et al., 2024). We leave this possible direction to future work.

## 6 Conclusions

We introduce inference-time *hyper-scaling*: by compressing the key–value cache of Transformer LLMs via sparse attention, we improve downstream reasoning accuracy by enabling longer token sequences or more parallel sequences at the same compute budget—in terms of latency or memory—compared to the original LLM. A fundamental requirement of inference-time hyper-scaling is to increase efficiency without sacrificing accuracy. To achieve this, we propose Dynamic Memory Sparsification (DMS), a novel, trainable KV cache reduction method that delays eviction decisions, while remaining highly data-efficient. Empirically, we observe large gains on benchmarks involving advanced math, scientific problems, and coding, demonstrating the effectiveness of hyper-scaling. Overall, our approach provides an inexpensive strategy for converting LLMs into more effective reasoners, pushing inference-time scaling to new frontiers.

## Acknowledgments

This work is supported by the ERC Starting Grant AToM-FM (101222956) awarded to Edoardo M. Ponti. The authors would like to thank Marcin Chochowski, David Tarjan, and Andrzej Sułecki for helpful discussions, Szymon Migacz for his assistance with the computing infrastructure, as well as Przemysław Strzelczyk, Krzysztof Pawelec, Daniel Korzekwa, Alex Fit-Florea, and Michael Lightstone for support in releasing this paper.

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

# A  Limitations, Future Work and Impact

**Larger Model Sizes, Longer Contexts, and Higher Compression Ratios**    In this work, we focus on models ranging from 1B to 32B parameters, context lengths up to 32K tokens, and compression ratios up to $8\times$. Exploring even larger models, longer contexts, and higher compression ratios remains an exciting avenue for future research.

**Integration with Other Efficient Attention Mechanisms**    We demonstrated DMS with the standard multi-head attention mechanism used in Transformer-based models such as Llama and Qwen. Extending DMS to alternative attention variants, such as Multi-head Latent Attention (DeepSeek-AI, 2024) represents a promising direction for future investigation. Moreover, DMS compresses the KV cache, whereas Quest (Tang et al., 2024) selectively retrieves cache items. Hence, the two are orthogonal and could be combined to further push the Pareto frontier for inference time scaling.

**Broader Impact**    Our approach does not introduce novel risks. However, it may amplify existing concerns associated with large-scale reasoning models. For a detailed analysis of these risks, we refer readers to Zhou et al. (2025).

# B  Related Work for KV Cache Size Reduction

The challenge of KV cache reduction has garnered significant interest in recent years, with approaches falling into three main categories: attention sparsification, quantization, and decomposition. In addition to the sparse attention baselines considered in Section 2.2, Landmark Attention (Mohtashami and Jaggi, 2023) and Native Sparse Attention (Yuan et al., 2025) create representations for each KV cache chunk and retrieve only the most important chunks for attention computation, effectively reducing the amount of data transferred from the device HBM memory. Compared to these methods, DMS not only accelerates inference but also reduces memory load and allows for dynamically selecting different compression ratios across layers and heads based on the input. Moreover, DMS improves on other retrofitting methods, such as DMC (Nawrot et al., 2024), both in terms of data efficiency and downstream accuracy.

Another strategy for KV cache size reduction is quantization, exemplified by methods such as KIVI (Liu et al., 2024) and KVQuant (Hooper et al., 2024), which quantize keys per channel and values per token. Finally, KV cache reduction can be achieved via SVD-based decomposition. LoRC (Zhang et al., 2024) directly reduces the ranks of key and value matrices, whereas Eigen Attention (Saxena et al., 2024) moves the attention computation into a truncated space induced by SVD. While being less expressive than DMS as they assume uniform compression, both quantization and decomposition are orthogonal to DMS and can be potentially combined with it to further improve efficiency.

# C   Additional Details for Retrofitting

**DMS Implementation**   Unlike (Nawrot et al., 2024), which extracts $\alpha_t$ from key representations affecting all query heads in a group, we 'borrow' the first neuron from the first query head in each query group and use it to extract $\alpha_t$, eliminating the need for additional parameters while minimizing the impact on attention computation. This requires a short continued training, during which we gradually zero out the first dimension of the first query head in each group: $\mathbf{q}_{t,\text{first}}[0] \leftarrow \mathbf{q}_{t,\text{first}}[0] \times \left(1 - \frac{t}{n_t}\right)$, where $t$ denotes the current training step and $n_t = 2000$. After this initial stage, the models are ready for the main DMS retrofitting phase, where they learn to dynamically evict tokens. After we extract $\alpha_t$ from the first query head, we set $\mathbf{q}_{t,\text{first}}[0] = 0$ to avoid $\alpha_t$ influence on the result of attention calculation, while leaving other query heads in the group unaffected. We note that instead of borrowing the neuron from a query head, one could use a separate, trainable, zero-initialized projection from the hidden state to extract $\alpha_t$, effectively eliminating the need for continued training.

**Training Configuration**   We use a batch size of 1024 following the original Llama recipe (Touvron et al., 2023). Context lengths are set to 4096 tokens for Llama 3.2 1B Instruct and Llama 2 7B models, and 8192 tokens for Llama 3.1 8B and R1-distilled models to accommodate the longer sequences required by AIME and MATH-500 benchmarks. For Qwen3-8B we use 256 batch size and 32K context length.

**Default DMS Configuration**   Unless otherwise specified, all DMS models use delayed eviction with a sliding window of 256 tokens and increment the compression ratio by 1 every 100 training steps. We denote different DMS variants using the notation $\text{DMS}_{\text{win}=y}$, where $y$ represents the sliding window size. Unlike DMC (Nawrot et al., 2024), we omit the third fine-tuning phase (training with fixed compression ratio) as it provided negligible benefits for DMS.

**Infrastructure and Computational Requirements**   All models are trained on NVIDIA H100 GPUs using Megatron-LM (NVIDIA, 2024) in bfloat16 precision, with optimizer states stored in FP32. We provide details regarding the computational costs in Table 3.

Table 3: Cost of retrofitting the model from CR $i$ to CR $i+1$. We note that over the course of a single retrofitting run, the target CR is linearly increased from CR 1 to the target CR. As a result, a single run produces a family of models with different compression ratios.

| Model Family | #Params | CR $i \rightarrow$ CR $i+1$ | | | | | |
|---|---|---|---|---|---|---|---|
| | | BS | Context | Window | Steps | LR | GPU Hours |
| Llama 3 | 1B | 1024 | 4096 | 256 | 100 | 1e-5 | 10 |
| | 8B | | 8192 | 16 | 100 | 3e-5 | 100 |
| Qwen-R1 | 1.5B | 1024 | 8192 | 256 | 100 | 1e-5 | 30 |
| | 7B | | | | | 3e-5 | 75 |
| | 32B | | | | | 3e-5 | 345 |
| Qwen3 | 8B | 256 | 32768 | 512 | 100 | 3e-5 | 270 |

# D  Additional Inference-Time Scaling Results

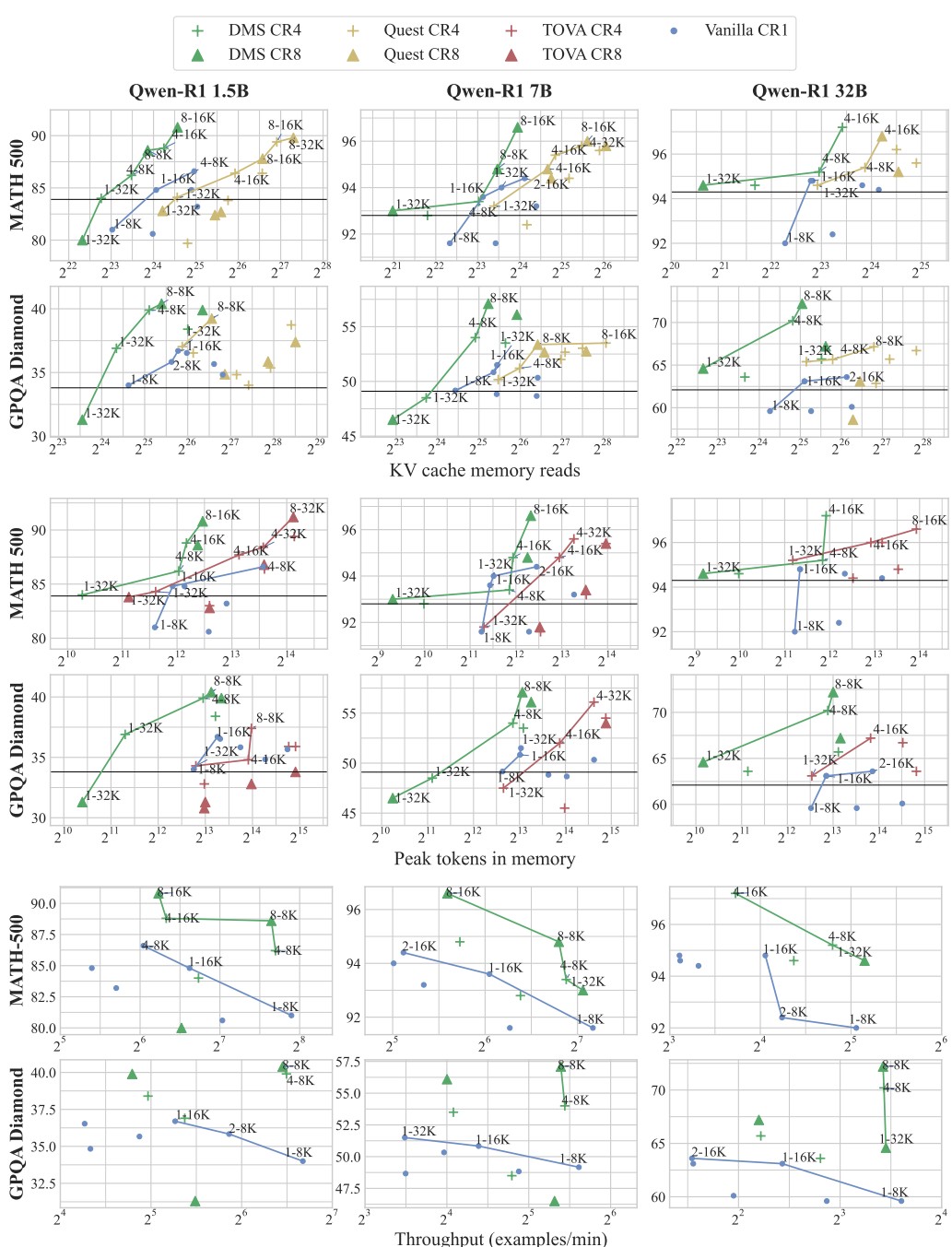

Figure 7: Inference-time scaling results calculated on MATH-500 and GPQA Diamond. The methods are compared in terms of accuracy on $y$-axis and efficiency metrics on $x$-axis: (**top**) KV cache memory reads, which serve as a proxy for attention compute; (**middle**) the maximum number of used tokens, as a proxy for memory load, and (**bottom**) throughput measured on NVIDIA H100 SXM GPU. For details, please refer to Figure 3 and Section 5.1.

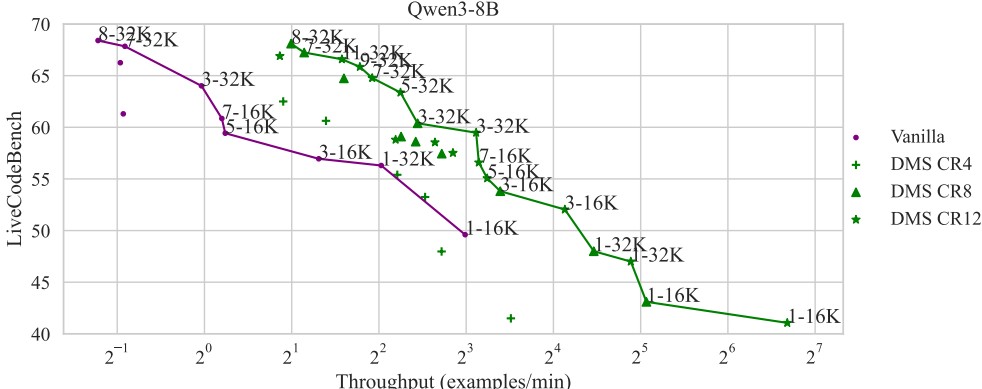

Figure 8: Throughput measured with maximum batch size that accommodates the particular inference-time scaling configuration. The plot compares DMS-enabled Qwen3-8B models at compression rations $4\times$, $8\times$ and $12\times$ to the vanilla model. It emerges from the plot that, through the combination of inference-time scaling and KV cache compression, DMS enables matching the vanilla results with up to $5\times$ higher throughput, effectively lowering the cost of serving the model in a multi-user environment.

# E    Training Data

For the Qwen-R1 models, we utilize logit distillation leveraging the OpenR1-Math-220k dataset. This dataset contains high-quality reasoning traces sampled from DeepSeek R1. To further enhance data quality, we apply a filtering step using Math-Verify (Kydlíček and Gandenberger, 2025), retaining only traces resulting in correct mathematical solutions.

For the Llama 3.2 1B Instruct model, the training corpus comprises two main components: (1) a carefully curated set of programming language examples covering languages such as Python, C, and C++, and (2) synthetic data generated by prompting the model. In particular, we utilize the Llama 3.2 1B Instruct model itself to produce completions for the one-dimensional linear algebra subset of the DeepMind mathematics dataset (Saxton et al., 2019), which follows the structured format:

> **Task format in one-dimensional linear algebra**
>
> ```
> Solve aX + b = cX + d for X.
> ```

> **Llama 3.2 1B prompt for generating responses**
>
> ```
> <|start_header_id|>system<|end_header_id|>
>
> Cutting Knowledge Date: December 2023
> Today Date: 23 July 2024
>
> You are a helpful
> ↪   assistant.<|eot_id|><|start_header_id|>user<|end_header_id|>
>
> Given the following problem, reason and give a final answer to the
> ↪   problem.
> Problem: Solve 5*b - 2355 = -50*b - 2740 for b.
> Your response should end with "The final answer is [answer]" where
> ↪   [answer] is the response to the
> ↪   problem.<|eot_id|><|start_header_id|>assistant<|end_header_id|>
> ```

In contrast with the data mixture for Qwen-R1 models, we do not perform correctness filtering on this synthetic, model-generated dataset.

# F    Additional Downstream Evaluations for DMC and DMS

In Table 4 we show that DMS can extrapolate beyond the retrofitting context length of 4K, whereas DMC may fail to do so. In Table 5, we show a comparison between Vanilla model, DMS, Quest, and DMC on Llama 2 7B.

Table 4: Needle in the Haystack and Variable Tracking results for 1B parameter Llama 3.2 Instruct model. We note that in contrast to DMC, DMS can extrapolate beyond the retrofitting context length. Note that on the heavily compressible Variable Tracking task, DMS achieves significantly higher scores than the vanilla model.

| Method/Task | NIAH | | | VT | | |
|---|---|---|---|---|---|---|
| Context | 3K | 4K | 8K | 3K | 4K | 8K |
| Vanilla | 99.4 | 96.4 | 97.2 | 61.4 | 55.8 | 41.2 |
| CR2 | | | | | | |
| TOVA | 62.8 | 65.2 | 75.0 | 56.0 | 56.2 | 49.8 |
| H2O | 29.0 | 34.0 | 37.0 | 25.6 | 27.4 | 21.4 |
| Quest | **99.2** | 95.8 | 97.4 | 60.0 | 53.0 | 36.4 |
| DMC | 99.0 | 0.0 | 0.0 | 62.4 | 0.0 | 0.0 |
| DMS$_{\text{win}=16}$ | 99.0 | **97.8** | **99.4** | **72.0** | **63.2** | **56.0** |
| CR3 | | | | | | |
| TOVA | 25.8 | 40.2 | 41.6 | 38.4 | 45.2 | 40.6 |
| H2O | 16.6 | 17.2 | 19.8 | 15.6 | 17.6 | 13.4 |
| Quest | 99.0 | **95.6** | **97.0** | 60.4 | 50.4 | 31.8 |
| DMC | 99.0 | 1.8 | 0.0 | 46.4 | 0.2 | 1.2 |
| DMS$_{\text{win}=16}$ | **99.2** | 93.6 | 24.2 | **76.2** | **69.2** | **58.8** |
| CR4 | | | | | | |
| TOVA | 16.8 | 28.0 | 26.4 | 31.4 | 33.8 | 30.2 |
| H2O | 9.4 | 13.4 | 12.8 | 11.8 | 12.6 | 11.0 |
| Quest | 98.4 | 95.8 | **97.6** | 57.4 | 49.6 | 32.4 |
| DMC | 97.0 | 0.0 | 0.0 | 48.6 | 4.0 | 0.8 |
| DMS$_{\text{win}=16}$ | **99.4** | **96.8** | 12.2 | **74.8** | **67.6** | **57.2** |

Table 5: Results for base Llama 2 7B parameter models. Both DMS and DMC were trained using LM-loss without logit distillation. Since these models are not instruction-tuned, we evaluate with 8-shot prompting on GSM8K, 5-shot on MMLU, 1-shot Needle in a Haystack, and zero-shot on ARC-Challenge and HellaSwag. (Nawrot et al., 2024).

| Method | ARC-C | GSM8K | HS | MMLU | NIAH |
|---|---|---|---|---|---|
| Vanilla | 45.6 | 14.9 | 75.5 | 45.4 | 100.00 |
| CR4 | | | | | |
| DMS$_{\text{win}=16}$ | 45.8 | 14.2 | 76.0 | 43.7 | 100.0 |
| Quest | 45.6 | 14.5 | 75.5 | 45.4 | 100.0 |
| DMC | 46.2 | 12.2 | 76.3 | 43.9 | 100.0 |
| CR8 | | | | | |
| DMS$_{\text{win}=16}$ | 46.2 | 10.5 | 76.4 | 40.2 | 60.0 |
| Quest | 45.6 | 11.6 | 75.5 | 45.4 | 100.0 |
| DMC | 44.7 | 10.0 | 75.4 | 41.7 | 100.0 |

Table 1 compares downstream performance at $2\times$, $3\times$, and $4\times$ compression ratios.[4] DMS stands out as the most robust method, achieving higher scores than both training-free and retrofitted baselines in most combinations of tasks and CRs, with Quest as a second-best contender. While in short-context tasks, DMS performance is close to the original LLM, in long-context tasks (such as NIAH and VT) DMS even surpasses it. Moreover, long-context performance provides evidence that—compared with DMC—DMS is more successful at extrapolating compression to lengths beyond those observed during retrofitting, albeit only up to a certain limit (see Appendix F). Among learned compression methods, DMC collapses quickly, likely due to its more challenging training objective amplified by the limited 1B model capacity.[5]

## G    Downstream Results Significance

We provide further analysis regarding the statistical significance and robustness of our experimental results. Specifically, we report standard errors for the Llama 3.2 1B Instruct models in Table 6, and quantify the average Pareto improvement in Tables 8 and 9. To precisely measure the Pareto improvement, we extract Pareto frontiers for DMS, the best KV cache reduction baseline, and the vanilla baseline from Figures 3 and 7 (top and middle). Then, for each task and model size, we identify the largest common budget interval $I$ shared by each pair of methods A and B, and compute the average improvement as:

$$\frac{\int_{x \in I} \left( A(x) - B(x) \right) dx}{|I|}$$

where $A(x)$ and $B(x)$ denote the best accuracy achieved by method A and B, respectively, at budget $x$. For budget values not explicitly measured, we employ linear interpolation.

Table 6: Results from Table 1 expanded with standard error as computed by Language Model Evaluation Harness (Gao et al., 2024).

| Method | ARC-C | GPQA | GSM8K | HS |
|---|---|---|---|---|
| Vanilla | $31.2_{\pm 1.4}$ | $25.0_{\pm 2.0}$ | $44.9_{\pm 1.4}$ | $43.4_{\pm 0.5}$ |
| CR2 | | | | |
| $DMS_{win=16}$ | $31.3_{\pm 1.4}$ | $25.7_{\pm 2.1}$ | $46.6_{\pm 1.4}$ | $43.3_{\pm 0.5}$ |
| TOVA | $29.6_{\pm 1.3}$ | $25.2_{\pm 2.1}$ | $45.0_{\pm 1.4}$ | $42.8_{\pm 0.5}$ |
| H2O | $31.1_{\pm 1.4}$ | $26.8_{\pm 2.1}$ | $44.0_{\pm 1.4}$ | $42.9_{\pm 0.5}$ |
| Quest | $31.2_{\pm 1.4}$ | $25.0_{\pm 2.0}$ | $45.1_{\pm 1.4}$ | $43.4_{\pm 0.5}$ |
| CR3 | | | | |
| $DMS_{win=16}$ | $31.1_{\pm 1.4}$ | $24.6_{\pm 2.0}$ | $45.5_{\pm 1.4}$ | $43.3_{\pm 0.5}$ |
| TOVA | $30.0_{\pm 1.3}$ | $23.7_{\pm 2.0}$ | $40.1_{\pm 1.4}$ | $42.5_{\pm 0.5}$ |
| H2O | $31.2_{\pm 1.4}$ | $24.3_{\pm 2.0}$ | $32.9_{\pm 1.3}$ | $42.1_{\pm 0.5}$ |
| Quest | $31.2_{\pm 1.4}$ | $25.0_{\pm 2.0}$ | $44.7_{\pm 1.4}$ | $43.4_{\pm 0.5}$ |
| CR4 | | | | |
| $DMS_{win=16}$ | $31.1_{\pm 1.4}$ | $24.3_{\pm 2.0}$ | $41.0_{\pm 1.4}$ | $43.4_{\pm 0.5}$ |
| TOVA | $29.0_{\pm 1.3}$ | $23.7_{\pm 2.0}$ | $20.2_{\pm 1.1}$ | $41.8_{\pm 0.5}$ |
| H2O | $27.5_{\pm 1.3}$ | $23.7_{\pm 2.0}$ | $14.7_{\pm 1.0}$ | $41.3_{\pm 0.5}$ |
| Quest | $31.2_{\pm 1.4}$ | $25.0_{\pm 2.0}$ | $39.9_{\pm 1.3}$ | $43.4_{\pm 0.5}$ |

---

[4]While H2O and TOVA are designed for long context tasks with large sliding windows, we consciously evaluate them with short sliding windows to meet the target CRs.

[5]Nevertheless, in Appendix F we show that, while still lagging behind, this collapse does not occur for shorter contexts and a larger non-GQA model.

Table 7: Throughput-Accuracy Pareto frontier difference. We use linear interpolation for the unknown values of the frontier.

| Method | AIME 24 | | | MATH 500 | | | GPQA Diamond | | | LiveCodeBench | | |
|---|---|---|---|---|---|---|---|---|---|---|---|---|
| | 1.5B | 7B | 32B | 1.5B | 7B | 32B | 1.5B | 7B | 32B | 1.5B | 7B | 32B |
| DMS vs Vanilla | −1.5 | 10.1 | 7.4 | 5.0 | 2.2 | 3.3 | 5.6 | 6.0 | 8.4 | 8.1 | 7.3 | 13.2 |

Table 8: KV Reads-Accuracy Pareto frontier difference. We use linear interpolation for the unknown values of the frontier. NA denotes that the projections of the Pareto frontiers on the budget axis are disjoint.

| Method | AIME 24 | | | MATH 500 | | | GPQA Diamond | | | LiveCodeBench | | |
|---|---|---|---|---|---|---|---|---|---|---|---|---|
| | 1.5B | 7B | 32B | 1.5B | 7B | 32B | 1.5B | 7B | 32B | 1.5B | 7B | 32B |
| DMS vs Vanilla | 10.6 | 15.0 | 12.0 | 4.2 | 1.0 | 1.6 | 4.8 | 4.1 | 8.6 | 7.3 | 7.9 | 9.7 |
| Quest vs Vanilla | −6.8 | 3.1 | 2.0 | −1.8 | −0.6 | NA | NA | NA | 2.3 | 2.5 | 5.0 | 3.8 |
| DMS vs Quest | 18.8 | 13.5 | 5.8 | 6.2 | 2.1 | 1.4 | NA | NA | NA | 4.9 | 3.4 | 5.6 |

Table 9: KV Memory Usage-Accuracy Pareto frontier difference. We use linear interpolation for the unknown values of the frontier. NA denotes that the projections of the Pareto frontiers on the budget axis are disjoint.

| Method | AIME 24 | | | MATH-500 | | | GPQA Diamond | | | LiveCodeBench | | |
|---|---|---|---|---|---|---|---|---|---|---|---|---|
| | 1.5B | 7B | 32B | 1.5B | 7B | 32B | 1.5B | 7B | 32B | 1.5B | 7B | 32B |
| DMS vs Vanilla | 17.3 | 15.7 | 14.6 | 3.4 | 0.5 | 1.6 | 4.9 | 4.2 | 8.0 | 7.4 | 8.5 | 16.7 |
| TOVA vs Vanilla | 5.3 | −0.2 | 2.6 | 1.3 | −1.2 | 1.8 | −1.1 | −2.1 | 2.1 | 3.8 | 3.3 | 4.9 |
| DMS vs TOVA | 9.6 | 15.6 | 8.1 | 2.3 | 1.8 | −0.1 | 5.6 | 6.5 | 6.3 | 4.0 | 6.0 | 11.1 |

Table 10: Results from Figure 3 and Figure 7 (top) specified max Length and Width=1 configurations. Those points allow for a direct comparison with Vanilla model.

| Task | Model | Size | CTX | Vanilla | DMS CR 4 | Quest CR4 |
|---|---|---|---|---|---|---|
| AIME 24 | Qwen-R1 | 1.5B | 32k | 30.0 | 30.0 | 26.7 |
| | | 7B | 32k | 53.3 | 53.3 | 55.5 |
| | | 32B | 32k | 70.0 | 73.3 | 73.3 |
| MATH 500 | Qwen-R1 | 1.5B | 32k | 84.8 | 84.0 | 84.1 |
| | | 7B | 32k | 94.0 | 92.8 | 93.2 |
| | | 32B | 32k | 94.8 | 94.6 | 94.6 |
| GPQA Diamond | Qwen-R1 | 1.5B | 32k | 36.5 | 36.9 | 37.0 |
| | | 7B | 32k | 51.5 | 48.5 | 50.2 |
| | | 32B | 32k | 63.1 | 63.6 | 65.4 |
| LiveCodeBench | Qwen-R1 | 1.5B | 16k | 17.3 | 17.0 | 15.8 |
| | | 7B | 16k | 35.9 | 34.4 | 33.1 |
| | | 32B | 16k | 57.0 | 54.8 | 54.5 |

Table 11: Results from Figure 3 and Figure 7 (middle) specified max Length and Width=1 configurations. Those points allow for a direct comparison with Vanilla model.

| Task | Model | Size | CTX | Vanilla | DMS CR4 | TOVA CR4 |
|------|-------|------|-----|---------|---------|----------|
| AIME 24 | Qwen-R1 | 1.5B | 32k | 30.0 | 30.0 | 30.0 |
| | | 7B | 32k | 53.3 | 53.3 | 46.7 |
| | | 32B | 32k | 70.0 | 73.3 | 70.0 |
| MATH 500 | Qwen-R1 | 1.5B | 32k | 84.8 | 84.0 | 84.3 |
| | | 7B | 32k | 94.0 | 92.8 | 91.8 |
| | | 32B | 32k | 94.8 | 94.6 | 95.2 |
| GPQA Diamond | Qwen-R1 | 1.5B | 32k | 36.5 | 36.9 | 34.3 |
| | | 7B | 32k | 51.5 | 48.5 | 47.5 |
| | | 32B | 32k | 63.1 | 63.6 | 63.1 |
| LiveCodeBench | Qwen-R1 | 1.5B | 16k | 17.3 | 17.0 | 14.9 |
| | | 7B | 16k | 35.9 | 34.4 | 30.7 |
| | | 32B | 16k | 57.0 | 54.8 | 51.1 |

Table 12: Results from Figure 3 and Figure 7 comparing DMS wit CR8 to Vanilla (CR1) on specified max Length and Width=1 configurations.

| Task | Model | Size | CTX | Vanilla | DMS CR8 |
|------|-------|------|-----|---------|---------|
| AIME 24 | Qwen-R1 | 1.5B | 32k | 30.0 | 23.3 |
| | | 7B | 32k | 53.3 | 50.0 |
| | | 32B | 32k | 70.0 | 73.3 |
| MATH 500 | Qwen-R1 | 1.5B | 32k | 84.8 | 80.0 |
| | | 7B | 32k | 94.0 | 93.0 |
| | | 32B | 32k | 94.8 | 94.6 |
| GPQA Diamond | Qwen-R1 | 1.5B | 32k | 36.5 | 31.3 |
| | | 7B | 32k | 51.5 | 46.5 |
| | | 32B | 32k | 63.1 | 64.6 |
| LiveCodeBench | Qwen-R1 | 1.5B | 16k | 17.3 | 16.1 |
| | | 7B | 16k | 35.9 | 33.4 |
| | | 32B | 16k | 57.0 | 51.7 |

# H Evaluation Details

## H.1 Implementation of TOVA, H2O, Quest, and DMC

For TOVA (Oren et al., 2024), H2O (Zhang et al., 2023a), and Quest (Tang et al., 2024), we calculate the KV-budget by summing the input length and the maximum generation length, then dividing by the compression ratio. For H2O, the KV-budget is evenly split between the recent cache and the heavy-hitter cache. During evaluation, memory-optimising methods such as TOVA and H2O first perform a standard prefill phase until the KV-budget is reached and subsequently switch to their respective memory-efficient mechanisms.

Quest (Tang et al., 2024), unlike TOVA, H2O, DMC, and DMS, does not reduce the KV cache memory footprint. Thus, following the authors' recommendations, we permit Quest to perform prefilling using full dense attention and set the block size to $\max(16, 2cr)$. This configuration provides Quest with an advantage over the other methods. Additionally, we employ a separate top-k for each query head, which can result in an increased number of memory transfers for Quest compared to DMS, DMC, TOVA, and H2O. However, the computational cost remains similar. We use this approach as Quest was originally designed for models without GQA, and we wanted to avoid a custom modification that could potentially degrade the performance. In plots regarding kv-cache memory reads we calculate the total number of different blocks retrieved from a single key-head. That is we assume optimal implementation that makes use of topk intersections between query heads and retrieves each block only once.

For DMC, we follow the implementation described in the original paper (Nawrot et al., 2024).

## H.2 Downstream Tasks

We evaluate all downstream tasks in a zero-shot setting, unless stated otherwise.

For Qwen3 results in Table 2, we evaluate the model using temperature$= 0.6$ and top-$p = 0.95$ with a sequence length limit of 131072 tokens. AIME 2024 results were averaged over 10 runs (different seeds) and MATH-500 over 3; MMLU-Pro uses micro-averaging.

For GSM8K (Cobbe et al., 2021), MMLU (Hendrycks et al., 2021a), ARC-Challenge (Clark et al., 2018), and HellaSwag (Zellers et al., 2019), we use the Language Model Evaluation Harness (Gao et al., 2024), version 0.4.3.

For Needle in a Haystack (NIAH) (Kamradt, 2023) and Variable Tracking (VT), we adopt the evaluation implementation provided by RULER (Hsieh et al., 2024) and use the context length defined in the retrofitting procedure. For NIAH, we utilize the essay version with a single needle, whereas for VT, we utilize the version with 40 variable chains and 0 hops, filled with repeating text.

For AIME24,[6] GPQA Diamond (Rein et al., 2024), and MATH-500 (Lightman et al., 2024), we evaluate models using the Search and Learn framework (version 0.1.0) (Snell et al., 2024; Beeching et al., 2024), with math-parsing functionality derived from MathVerify (version 1.0.0) (Kydlíček and Gandenberger, 2025). For LiveCodeBench we utilize the tasks from 2024-08-01 till 2025-01-31.

For few-shot tasks from Language Model Evaluation Harness we directly utilize the framework to provide the few-shot examples. For RULER (Hsieh et al., 2024), we use the example generator to sample few-shot examples. Below we present remaining prompts that were used during the evaluation, except the prompts to base models, which were set to unaltered task input, and prompts for zero-shot evaluation of instruction tuned models, which were set to task inputs wrapped with HuggingFace tokenizer chat template.[7]

For GSM8K zero-shot evaluation, we adopt the prompt from Meta.

---

[6] https://huggingface.co/datasets/HuggingFaceH4/aime_2024
[7] https://huggingface.co/docs/transformers/en/chat_templating

```
 GSM8K zero-shot prompt

 <|start_header_id|>system<|end_header_id|>

 Cutting Knowledge Date: December 2023
 Today Date: 23 July 2024

 You are a helpful
 ↪   assistant.<|eot_id|><|start_header_id|>user<|end_header_id|>

 Given the following problem, reason and give a final answer to the
 ↪   problem.
 Problem: ___problem_text___
 Your response should end with "The final answer is [answer]" where
 ↪   [answer] is the response to the
 ↪   problem.<|eot_id|><|start_header_id|>assistant<|end_header_id|>
```

For Qwen-R1 AIME 24, MATH-500 and GPQA $\Diamond$ we adopt the prompts from Open-R1 repository[8].

```
 AIME 24 and MATH-500 prompts

 <|User|>Solve the following math problem efficiently and clearly.
 ↪   The last line of your response should be of the following format:
 ↪   'Therefore, the final answer is: $\boxed{ANSWER}$. I hope it is
 ↪   correct' (without quotes) where ANSWER is just the final number
 ↪   or expression that solves the problem. Think step by step before
 ↪   answering.

 ___problem_text___<|Assistant|><think>
```

```
 GPQA Diamond prompt

 <|User|>Answer the following multiple choice question. The last line
 ↪   of your response should be of the following format: 'Answer:
 ↪   $LETTER' (without quotes) where LETTER is one of ABCD. Think step
 ↪   by step before answering.

 ___problem_text___<|Assistant|><think>
```

For coding tasks we utilize the following prompt adopted from LiveCodeBench(Jain et al., 2025)
DeepSeek-R1 setting:

---

[8]https://github.com/huggingface/open-r1

# I  Influence of KV Cache on Inference Latency

In this section, we provide a simplified analysis estimating the proportion of inference latency introduced by reading from the key–value cache to the entire latency of the step, during a single auto-regressive inference step of an LLM on a GPU. Our calculations are based on the architecture of the Llama 3 model family. Specifically, we derive sample equations for Llama 3.1 8B, parameters of which are listed below.

| Parameter | Value | Description |
|---|---|---|
| $n$ | 32 | Number of layers |
| $d$ | 4096 | Hidden dimension |
| $d_{ff}$ | 14336 | Internal dimension of the MLP layers |
| $d_{kv}$ | 1024 | Dimension of the Key/Value sequences |
| $V$ | 128256 | Vocabulary size |

The estimates can be expressed in terms of batch size $B$ and sequence length $L$, which determine the number of tokens in the KV cache. The number of floating-point operations (FLOPs) can be approximated as

$$\text{FLOPS}(B, L) \approx nB \left(6dd_{ff} + 4d^2 + 4dd_{kv} + 4dL\right) + 2BdV. \tag{2}$$

This calculation considers only major matrix-vector multiplications (assuming two FLOPs per multiply-accumulate operation), omitting minor operations such as normalization and pointwise non-linearities.

Similarly, we estimate the number of reads from the High Bandwidth Memory as:

$$\text{Reads}(B, L) \approx n \left(6dd_{ff} + 4d^2 + 4dd_{ff} + 4BLd_{kv}\right) + 2dV, \tag{3}$$

assuming 2 bytes per parameter (16-bit precision). Note that only the KV cache ($4nBLd_{kv}$) scales with batch size and sequence length. As a sanity check, we confirm that $\text{Reads}(1, 0)/2 \approx 7.5B$ approximate the model's parameter count (without $0.5B$ for the input embedding table, which does not have to be entirely read during an inference step). Finally, the approximations for Llama 3.1 8B have the following form:

$$\text{FLOPS}(B, L) \approx 1.45 \cdot 10^9 B + 5.24 \cdot 10^5 BL \tag{4}$$

$$\text{Reads}(B, L) \approx 1.50 \cdot 10^{10} + 1.31 \cdot 10^5 BL. \tag{5}$$

For the remaining calculations, we use the peak performance values for NVIDIA H100 SXM (https://www.nvidia.com/en-us/data-center/h100/) for 16-bit calculations without 2:4 sparsity:

| | |
|---|---|
| BFLOAT16 Tensor Core performance | 989.5 TFLOPS |
| GPU Memory bandwith | 3.35 TB/s |

Since memory reads are significantly slower than computations, the latency contribution from KV cache reads ($1.31 \times 10^5 BL$ term) dominates at larger sequence lengths and batch sizes. Thus, KV cache size is a critical factor in inference latency for long sequences.

The inference latency per step can be approximated as

$$\max \left( \frac{\text{FLOPS}(B, L)}{989.5 \, \text{TFLOPS}}, \frac{\text{Reads}(B, L)}{3.35 \, \text{TB/s}} \right), \tag{6}$$

assuming ideal overlap between computation and memory operations. Approximating KV cache reads as $4nBLd_{kv}$ and following the same calculations for other Llama and Qwen models, we visualize the fraction of latency contributed by KV cache reads to the latency of entire inference steps (Figure 9).

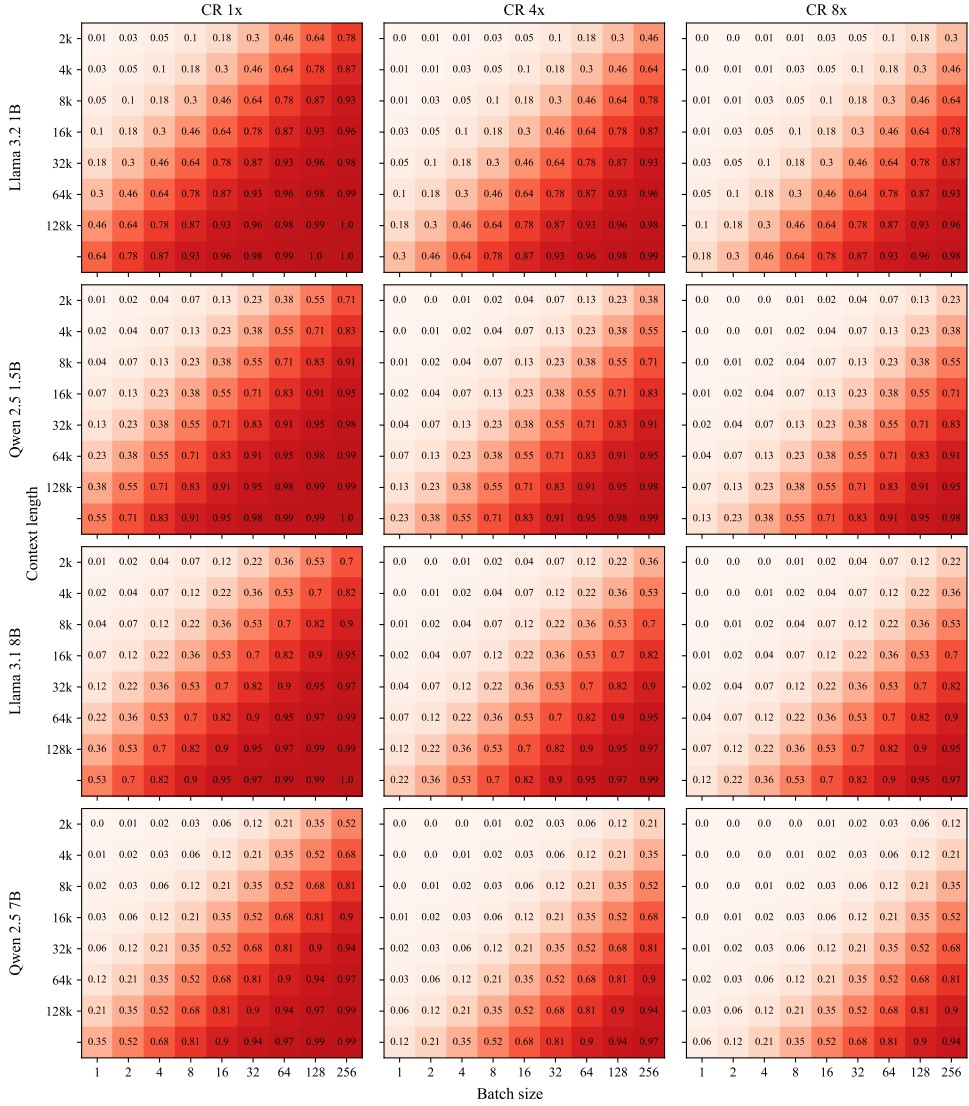

Figure 9: Percentage of total latency attributed to KV cache reads. Those reads clearly dominate latency as batch size and sequence length increase. When the KV cache is compressed (CR 4× and 8×), more tokens can be accommodated before the latency of reading the KV cache becomes an issue.

