# OpenReview forum: "Inference-Time Hyper-Scaling with KV Cache Compression"
_NeurIPS.cc/2025/Conference — NeurIPS 2025 poster_

### Official Review · Reviewer_eA55 · 2025-06-30

**Clarity:** 3
**Significance:** 3
**Originality:** 2
**Rating:** 5
**Confidence:** 2

**Summary:**

This paper introduces Dynamic Memory Sparsification (DMS), which closely mirrors Dynamic Memory Compression of Nawrot et al. 2024, except that it decides on evicting tokens from the KV cache instead of accumulating tokens on top of existing tokens. The approach yields improvements for the Pareto-frontier between efficiency and quality on a broad set of reasoning-intensive tasks.

**Questions:**

- Figures 2 and 3 are a bit crowded with the W-L-CR representation. I would encourage the authors to tidy up those plot for easier readability.

- How would the right subfigure of Figure 4 look like with a 1B training token budget (as in the DMC paper?)

**Ethical Concerns:**

["NO or VERY MINOR ethics concerns only"]

**Final Justification:**

This paper builds naturally on top of Dynamic Memory Compression, except for Sparsification, without any significant weaknesses in my opinion, although my familiarity with this line of work is limited. I maintain a positive assessment of this paper, as a valuable continuation of DMC.

**Limitations:**

The authors don't have a devoted discussion of their limitations. I would encourage them to contextualize the limitations of DMS in the limitations of DMC (e.g. alternatives to the DMC training procedure adopted here).

**Quality:**

3

**Strengths And Weaknesses:**

### Strengths

- The paper builds in an intuitive way on the prior success of DMC to create a simpler method of DMS. The proposal is an intuitive one, mirroring the merits of DMC.

- Delayed eviction is a simple and creative technique that is shown to effectively reduce KV cache size without quality degradation.

- The evaluation is strong, as DMS is shown to be on the Pareto frontier of inference-time scaling quality with respect to efficiency.

### Weaknesses

- The inference-time scaling configurations of the form W-L-CR are presented without much motivation or hypotheses as to which ones might be expected to yield the best results and why. That would be an useful study/ablation to include.

- The fact that the number of training tokens is 1B is not insignificant — does it fail with fewer training tokens? If so, this method is a relatively expensive one.

---

> ### Author Rebuttal · Authors · 2025-07-31
>
> We appreciate Reviewer eA55’s time and effort in reviewing our work and for the helpful suggestions they offered.
>
> **W1: Which L-W-CR setups yield the best results and why?**
>
> In our manuscript, we hint at the fact that sequential scaling rather than parallel scaling brings more gains (L240); however, we also remark that often hybrid strategies employing parallel *and* sequential scaling lie on the Pareto frontier. For CR, the key trade-off is whether the accuracy gains from scale outweigh accuracy losses due to sparsification. In our main experiments, CR 4x consistently achieves the best balance. Post-submission, we improved performance further using win=256, shifting the optimal CR to 8x - 10x. Hence, the optimal CR could be pushed even further by developing better retrofitting strategies and model configurations that do not degrade the model’s quality.
>
> **W2: Does the method fail with fewer than 1B training tokens?**
>
> DMS works even on small numbers of training tokens, as we demonstrate below in the answer to Question 2. However, longer retrofitting may reduce the variability of the results. In our experiments after the submission, we found that enlarging the sliding window also helps to stabilise the training and reduce variability, which can be seen in Figure 4 (right). With win=256, we were able to improve the results of R1-Qwen models shown in Figure 5 with only 100 steps for every CR increment instead of 3000 steps. With 100 steps for every CR increment at 8k context and batch size 1024, retrofitting requires 839M tokens per CR, and we have not attempted to optimise these numbers further. Please note that DMC uses 24B–168B tokens depending on CR; their 1B number applies to the neuron decay phase, not full adaptation.
>
> **Q1: Figures 2 and 3 are a bit crowded with the L-W-CR labels**
>
> We agree and will revise the plots to improve clarity: adjusting label positions, plot layout, font size, and potentially removing less important labels from non-Pareto points.
>
> **Q2: How would the right subfigure of Figure 4 look like with a 1B training token budget (as in the DMC paper?)**
>
> Reducing the amount of training data increases the variability of results, especially at small sliding window and model sizes. We have run additional experiments retrofitting Llama 3.2 1B Instruct with only 50 steps per CR, achieving CR 4x with ~630M training tokens. We ran 3 seeds for different window sizes (16, 64, 256). Considering that the vanilla model achieves 47.0 on GSM8K, the additional data-efficient GSM8K scores are:
>
> | Model                    | CR 2x | CR 3x | CR 4x |
> |--------------------------|------:|------:|------:|
> | DMS (win=16, seed 1)     | 43.4  | 44.5  | 28.7  |
> | DMS (win=16, seed 2)     | 41.8  | 41.9  | 39.3  |
> | DMS (win=16, seed 3)     | 41.8  | 33.1  | 35.6  |
> | DMS (win=64, seed 1)     | 41.7  | 43.1  | 40.3  |
> | DMS (win=64, seed 2)     | 43.9  | 41.8  | 41.8  |
> | DMS (win=64, seed 3)     | 42.4  | 35.9  | 42.1  |
> | DMS (win=256, seed 1)    | 43.9  | 45.6  | 43.9  |
> | DMS (win=256, seed 2)    | 45.1  | 43.9  | 44.9  |
> | DMS (win=256, seed 3)    | 43.9  | 44.7  | 43.3  |
>
> While DMS incurs some degradation due to the small model size and the extremely fast training regime, we note that a larger window size significantly stabilises training and mostly preserves the original performance.
>
> **A discussion of limitations is missing**
>
> We discuss the limitations of our work in Appendix A. We will also expand this in the Related Work section to highlight which limitations are inherited from DMC and which choices in the DMS training procedure were able to overcome some of the limitations of DMC, such as being data-hungry, more cumbersome to implement, being more prone to lead to performance degradation etc.

---

> > ### Comment · Reviewer_eA55 · 2025-08-04
> >
> > Thank you for your reply. The results provided in the rebuttal alleviate my concern about a potentially expensive training budget.
> >
> > I retain my positive assessment of this paper.

---

### Official Review · Reviewer_2bQ1 · 2025-07-01

**Clarity:** 3
**Significance:** 3
**Originality:** 3
**Rating:** 4
**Confidence:** 3

**Summary:**

This paper addresses the practical bottleneck of inference-time scaling in Transformer-based LLMs caused by the rapid growth of the key-value (KV) cache, particularly during reasoning-heavy tasks requiring long or multiple generation threads. The authors propose Dynamic Memory Sparsification (DMS), a method for compressing the KV cache that delays token eviction rather than immediately dropping tokens, thereby allowing models to preserve more critical information with less destructive context loss. DMS is positioned as a lightweight and sample-efficient retrofitting approach compared to prior methods, notably Dynamic Memory Compression (DMC) and existing training-free sparsification strategies. Through experiments on various LLMs (including DeepSeek R1-distilled and Llama 3.2 Instruct models) and multiple reasoning benchmarks (AIME 24, MATH 500, GPQA Diamond), the method demonstrates measurable improvements on the Pareto frontier of accuracy vs. efficiency (runtime and memory), outperforming existing KV cache compression techniques at similar or higher compression ratios.

**Questions:**

Hardware Efficiency Details: Can the authors provide more quantitative results for end-to-end inference metrics (e.g., wall-clock latency, throughput on mainstream GPU/TPU hardware) alongside the proxy measures (KV reads/peaks)? This will help practitioners assess real-world impact.

Generalization Beyond Reasoning Tasks: Have the authors considered, or do preliminary experiments support, the efficacy of DMS on common NLP tasks such as summarization, translation, dialogue, retrieval, or code completion? Would the benefits persist under very different prompting or sequence structures?

Potential Failure Cases: Are there identified or observed types of prompts or inputs where DMS harms or degrades model reasoning in an unexpected manner? E.g., extremely long-range dependencies, highly nonlocal attention patterns, or adversarial token input.

**Ethical Concerns:**

["NO or VERY MINOR ethics concerns only"]

**Final Justification:**

The authors have addressed the main concerns effectively. First, on task coverage, they added experiments on LiveCodeBench (code generation) and QASPER (long-context QA), showing that DMS maintains performance beyond long reasoning tasks. Second, on hardware-level speed-ups, they provided H100 latency benchmarks and clarified that gains appear primarily at large batch and sequence sizes; while the results are based on unoptimized research code, the trends are credible. For pathological inputs, they demonstrated strong results on NIAH and VT tasks involving long dependencies and non-local attention, though more systematic adversarial testing remains for future work. Regarding generalization, their explanation—supported by literature—on how small learning rates and short runs of logit distillation mitigate overfitting risks is convincing. Remaining limitations include broader NLP coverage, absence of adversarial stress tests, and latency validation on optimized inference engines. Overall, the additional experiments and clarifications substantiate the core contribution, and the remaining issues are natural extensions rather than fundamental flaws; I therefore maintain a positive recommendation.

**Limitations:**

yes

**Quality:**

3

**Strengths And Weaknesses:**

Strengths

The paper addresses a pressing challenge for deploying large language models (LLMs) at scale—specifically, the memory and runtime bottlenecks associated with attention computation via the key-value (KV) cache. This problem is increasingly relevant as LLM adoption grows, and existing methods struggle to balance efficiency and quality. The proposed method, DMS, is well-articulated and clearly distinguished from prior work. Its central idea—delayed eviction through a sliding window—represents a simple yet well-substantiated improvement over both immediate-dropping methods (e.g., TOVA, H2O) and merging strategies like DMC.

The empirical evaluation is thorough, covering a diverse set of LLM architectures including Qwen-R1-distilled (1.5B, 7B, 32B) and Llama 3.2 1B Instruct. Benchmarks span several challenging, reasoning-intensive datasets such as MATH 500, AIME 24, and GPQA Diamond. Results are well-presented in figures and tables: Figures 2 and 3 clearly illustrate trade-off curves (Pareto frontiers) between KV cache reads, peak tokens, and performance. DMS consistently outperforms alternatives in the accuracy-memory/runtime trade-off, particularly for larger models. Table 1 shows that DMS maintains high accuracy across datasets like GSM8K, MMLU, HellaSwag, NIAH, and VT even at 4× compression, where baseline methods degrade sharply.

Section 5.2 and Figure 4 offer a solid ablation and design analysis, exploring the impact of sliding window size, delayed versus immediate eviction, and the training token budget. These experiments yield practical guidance for tuning hyperparameters and understanding the cost of retrofitting. The methodology is presented clearly, with readable equations and sufficient implementation details to support reproducibility for expert audiences. The claims are backed by extensive empirical results. Section 6 effectively contextualizes DMS among contemporary techniques such as sparse attention, quantization, and low-rank approximations, and emphasizes its orthogonality or complementarity to those approaches.

Weaknesses

Despite its strengths, the paper’s evaluation lacks breadth in terms of task coverage. It focuses almost exclusively on complex reasoning tasks and does not explore standard NLP applications such as summarization, dialogue, or code generation, making the generality of DMS uncertain. Furthermore, although the work is motivated by memory and runtime limitations, it provides little hardware-level evidence. There are no direct reports on wall-clock speedups, VRAM savings, or inference latency and throughput on real devices, which would better contextualize the practical benefits of DMS.

The paper also omits analysis of failure modes. There is no exploration of how the method behaves on pathological prompts, out-of-distribution inputs, or inputs with irregular attention patterns—factors that could significantly affect robustness. Lastly, the reliance on logit distillation with sampled data for retrofitting raises concerns about generalization. It remains unclear how well the approach transfers to domains with significant distribution shifts or task variations.

---

> ### Author Rebuttal · Authors · 2025-07-31
>
> We thank Reviewer 2bQ1 for taking the time to read our paper and for providing thoughtful and constructive feedback.
>
> **W1 (a) and Q2: Broader representation of NLP tasks in the evaluation**
>
> We appreciate your suggestion to include a broader set of tasks! We note that DMS is primarily designed for compressing the KV cache during the auto-regressive decoding phase, rather than the parallel prefilling phase. Hence, the focus of our evaluation is primarily on long-generation/reasoning tasks rather than long-context tasks. Among the tasks you suggested, a prime example of the former is code generation: hence, we ran additional experiments for R1-Qwen 2.5 models on LiveCodeBench, trained with win=256 and 100 steps per CR. For parallel scaling, we report pass@all, namely whether any of the generated threads passed the tests for a given problem.
>
> | Model             | L-W-CR  | 1.5B Score | 1.5B KV Reads | 7B Score | 7B KV Reads | 32B Score | 32B KV Reads |
> |------------------|--------:|-----------:|--------------:|---------:|------------:|----------:|-------------:|
> | Vanilla          | 16-1-1  | 17.30      | 8.50E+07      | 35.90    | 7.21E+07    | 57.00     | 6.04E+07     |
> | DMS (win=256)    | 16-1-4  | 16.42      | 2.01E+07      | 34.62    | 2.04E+07    | 53.60     | 1.82E+07     |
> | DMS (win=256)    | 16-1-8  | 15.04      | 1.18E+07      | 32.61    | 1.20E+07    | 54.46     | 1.20E+07     |
> | Quest            | 16-1-4  | 16.96      | 5.95E+07      | 33.74    | 5.05E+07    | 54.42     | 4.54E+07     |
> | Quest            | 16-1-8  | 16.68      | 3.98E+07      | 32.02    | 3.50E+07    | 54.20     | 3.25E+07     |
> | TOVA             | 16-1-4  | 16.04      | 4.14E+07      | 32.22    | 3.71E+07    | 49.54     | 3.35E+07     |
> | DMS (win=256)    | 16-4-4  | 24.80      | 8.07E+07      | 44.90    | 8.17E+07    | 63.80     | 7.29E+07     |
> | Quest            | 16-4-4  | 25.40      | 2.37E+08      | 48.00    | 2.02E+08    | 64.10     | 1.81E+08     |
> | TOVA             | 16-4-4  | 25.10      | 1.66E+08      | 43.00    | 1.48E+08    | 58.50     | 1.34E+08     |
> | DMS (win=256)    | 16-8-8  | 26.60      | 9.44E+07      | 47.70    | 9.58E+07    | 70.00     | 7.49E+07     |
>
> Similar to other long-generation tasks, we find that hyper-scaling with DMS yields vastly improved performance for the same compute budget, measured as cumulative memory reads.
>
> In addition, we attach the results of Llama 3.1 8B on the QASPER benchmark (win=256, 200 steps per CR) for QA on scientific research papers. We find that DMS mostly maintains performance on this long-context NLP task.
>
> | Model    | Llama 3.1 8B |
> |----------|--------------|
> | Vanilla  |         33.1 |
> | DMS 8x   |         32.9 |
>
> While our focus is on long-sequence reasoning, these results indicate DMS maintains performance on other NLP tasks. We agree that broader evaluations would be valuable and are a natural extension of this work.
>
> **W1 (b) and Q1: Little hardware-level evidence for speed-ups, no reports of measurements**
>
> **We provide latency measurements during generation using the Megatron-LM framework**. We run our measurements on H100 GPUs in bfloat16 precision using FlashAttention2 and the flash_attn_with_kvcache routine for both vanilla and DMS. We note that we use research code for DMS that has not been optimised (e.g., through fusion of multiple operations), resulting in a visible latency overhead. In addition, we would like to note that for small to moderate token batch sizes, KV cache size does not drive the majority of latency (please refer to Appendix H and Figure 6 for details). However, a practical inference scaling scenario of serving multiple users at a time, requires substantial token batch sizes. In addition, Qwen 2.5 models have smaller KV cache sizes than the Llama 3 family or the newer Qwen 3 family, when measured in relation to the number of model parameters. Finally, we remark that our Megatron-LM framework setup is designed for fast inference, albeit we could not optimise it to the point of matching state-of-the-art inference engines due to time constraints of the response period.
>
> For latency benchmarks, we measure the generation of reasoning chains for AIME24. The notation “Lat @ X” denotes the latency when generating a token at position X. From the results below, it emerges that even for moderate batch sizes (such as 64 and 128) and sequence lengths (>-8k), CR4 and CR8 decrease latency considerably. The reason why there is not a direct correspondence between the reduction in memory reads (due to higher CR) and reductions in latency is that their dependency emerges only at large token batch and sequence sizes. We refer to Appendix H for an explanation with heatmaps. Overall, **these findings suggest that DMS results in effective latency reductions due to decreased KV cache traffic in regimes such as inference-time scaling (parallel or sequential)**. In turn, this enhances reasoning quality by allowing for a larger effective token budget for the same wall-clock runtime (while also reducing memory load).
>
> | Model | Batch size | Lat @ 4k | Lat @ 8k | Lat @ 12k | Lat @ 16k |
> |-|-|-:|-:|-:|-:|
> | R1-Qwen 1.5B Vanilla | 16 | 19.1 ms | 19.3 m | 18.9 ms | 18.7 ms |
> | R1-Qwen 1.5B DMS 4x | 16 | 27.5 ms | 26.1 ms | 26.2 ms | 26.3 ms |
> | R1-Qwen 1.5B DMS 8x | 16 | 26.6 ms | 26.3 ms | 26.4 ms | 25.5 ms |
> | R1-Qwen 1.5B Vanilla | 64 | 21.9 ms | 39.0 ms | 40.7 ms | 41.1 ms |
> | R1-Qwen 1.5B DMS 4x | 64 | 26.2 ms | 26.3 ms | 25.9 ms | 26.4 ms |
> | R1-Qwen 1.5B DMS 8x | 64 | 25.5 ms | 25.2 ms | 24.9 ms | 24.9 ms |
> | R1-Qwen 1.5B Vanilla | 128 | 23.1 ms | 41.0 ms | 58.4 ms | 76.0 ms |
> | R1-Qwen 1.5B DMS 4x | 128 | 26.0 ms | 25.8 ms | 26.2 ms | 26.1 ms |
> | R1-Qwen 1.5B DMS 8x | 128 | 26.7 ms | 26.1 ms | 26.1 ms | 25.9 ms |
>
> | Model | Batch size | Lat @ 4k | Lat @ 8k | Lat @ 12k | Lat @ 16k |
> |-|-:|-:|-:|-:|-:|
> | R1-Qwen 7B Vanilla | 16 | 24.9 ms | 25.7 ms | 26.4 ms | 27.1 ms |
> | R1-Qwen 7B DMS 4× | 16 | 25.9 ms | 25.8 ms | 25.8 ms | 26.3 ms |
> | R1-Qwen 7B DMS 8× | 16 | 26.7 ms | 27.2 ms | 27.4 ms | 26.5 ms |
> | R1-Qwen 7B Vanilla | 32 | 25.8 ms | 42.8 ms | 44.4 ms | 45.0 ms |
> | R1-Qwen 7B DMS 4× | 32 | 25.7 ms | 25.9 ms | 25.9 ms | 25.6 ms |
> | R1-Qwen 7B DMS 8× | 32 | 26.0 ms | 26.9 ms | 26.9 ms | 27.1 ms |
> | R1-Qwen 7B Vanilla | 64 | 26.8 ms | 44.5 ms | 62.2 ms | 79.7 ms |
> | R1-Qwen 7B DMS 4× | 64 | 26.1 ms | 26.8 ms | 26.8 ms | 26.4 ms |
> | R1-Qwen 7B DMS 8× | 64 | 26.2 ms | 27.0 ms | 26.8 ms | 26.0 ms |
>
> We will include latency plots for all model sizes in the final version of our paper.
>
> **W2 (a) and Q3: Exploration of how the method behaves on pathological prompts**
>
> The Needle-in-a-Haystack and Variable Tracking (VT) tasks (Table 1 in the manuscript; Tables 5 and 7 in Appendix D) consist of synthetically constructed inputs, and DMS models perform well on both (often exceeding the original model’s performance). In particular, the VT task measures the ability to track long-range dependencies, and NIAH attention patterns are highly non-local. As our models are retrofitted without access to private pre-training and post-training data, we cannot confirm whether the inputs for these two tasks are OOD, although there is a high chance that this is the case. Searching for adversarial inputs is interesting, but beyond our core focus on compressing KV caches for long reasoning chains, where we assume that the prompt is well-formatted.
>
> **W2 (b) Logit distillation with sampled data raises concerns about generalization. How well the approach transfers to domains with significant distribution shifts or task variations?**
>
> Logit distillation (model matching) mitigates domain mismatch more effectively than language modelling loss (data matching) when retrofitting LLMs [1, 2]. In addition, DMS performs well under short retrofitting runs (e.g., 100 steps per CR in the LiveCodeBench experiments above), which involve considerably less data than used for pre-training. Combined with small learning rates, this suggests minimal risk of overfitting to the distillation data and altering the model behaviour.
>
> [1] S.T. Sreenivas, S. Muralidharan, R. Joshi, M. Chochowski, A.S. Mahabaleshwarkar, G. Shen, J. Zeng, Z. Chen, Y. Suhara, S. Diao, C. Yu, W.-C. Chen, H. Ross, O. Olabiyi, A. Aithal, O. Kuchaiev, D. Korzekwa, P. Molchanov, M. Patwary, M. Shoeybi, J. Kautz, B. Catanzaro. LLM Pruning and Distillation in Practice: The Minitron Approach, 2024.
> [2] B. Minixhofer, I. Vulić, E. M. Ponti. Universal Cross-Tokenizer Distillation via Approximate Likelihood Matching. arXiv preprint arXiv:2503.20083, 2025.

---

> > ### Comment · Reviewer_2bQ1 · 2025-08-06
> >
> > I thank the authors for their thorough rebuttal. The additional explanations and results have successfully addressed my previous concerns. Therefore, I stand by my positive assessment of this manuscript.

---

### Official Review · Reviewer_BS8X · 2025-07-02

**Clarity:** 2
**Significance:** 3
**Originality:** 3
**Rating:** 4
**Confidence:** 4

**Summary:**

This paper investigate how KV cache compression facilitates inference-time scaling by allowing more reasoning samples or longer reasoning length. Moreover, it introduces Dynamic Memory Sparsification (DMS), a KV cache eviction method through post-training retrofitting, utilizing logits distillation during training and delayed eviction during inference. Experiments show that DMS improves performance-efficiency Parato frontiers for KV compression methods on reasoning tasks, and maintains near-lossless performance on other tasks.

**Questions:**

The writing of this paper requires major improvements. The “Logit Distillation and Retrofitting” and zeroing out of $q\[0\]$ are only introduced in the experimental setup section, making the Sec. 3 hard to understand (why $q\[0\]$ can predict the eviction decision).

The notation in Line 149 is also ambiguous: L, H, T denotes both the numbers and the sets. And is the total loss simply summation of distillation loss and auxiliary loss, with no coefficients needed?

In Line 156, it is not clear that why the additive attention is never materialized.

In Line 192, it seems that only one $q\[0\]$ is retrofitted for a GQA group, so are the eviction decisions consistent for all heads within a group? This requires more specific demonstration and notation.

In Figure 4 right, for win=16 (the preferred configuration), reducing the number of training tokens incurs significant drops in GSM8K scores, why the authors conclude that the model is robust (Line 259).

**Ethical Concerns:**

["NO or VERY MINOR ethics concerns only"]

**Final Justification:**

The authors have addressed my concerns and I will keep my original score.

**Limitations:**

For training or retrofitting, the essential issue of DMS is independent retrofitting for different compression rates. Moreover, the amount of retrofitting data increases for higher compression rates. These can result in high retrofitting costs, hindering its practical deployment.

For inference, DMS uses KV cache token reads as the metric for speed-efficiency. While it is reasonable to some extent, it would be more convincing to show practical latency speedups, which can cover the overhead of DMS eviction. The authors mentioned their efficiency considerations such as overwriting and PageAttention (Line 166, 168), while simply using KV cache token reads cannot demonstrate such considerations.

**Paper Formatting Concerns:**

The format instruction specified that only the first letter should be capital in titles of sections and paragraphs. This paper does not follow this requirement.

**Quality:**

3

**Strengths And Weaknesses:**

Strengths:
- Important problem: inference-time scaling is an emerging trend, while the memory and efficiency issues caused by large KV cache of generated tokens are rarely studied.
- The sparsification by retrofitting through logits distillation is novel and effective.
- Extensive experiments demonstrate that DMS improves the Pareto frontiers of KV cache compression on reasoning tasks.

Weaknesses:
- The writing of this paper is confusing, with some critical details unclear. See more in the Questions part.
- The costs of retrofitting are not mentioned. Moreover, DMS requires independent retrofitting for different compression rates, and the amount of retrofitting data increases for higher compression rates, which might hinder its practical deployment.
- Practical evaluation on efficiency is not provided.

---

> ### Author Rebuttal · Authors · 2025-07-31
>
> We appreciate Reviewer BS8X’s careful reading and helpful comments, which we address below.
>
> **W1: Improvements in writing**
>
> We will revise the manuscript to correct the title casing, resolve issues with the notation, and reorder the introduction of $q[0]$ (see below). We will also double-check the format instructions to rule out any other inconsistencies.
>
> **W2: Does DMS require independent retrofitting for different compression ratios?**
>
> Fortunately, this is not the case! A single run with a gradually increasing CR yields multiple checkpoints for various compression levels. All our experiments are based on this setup. This enables a more efficient use of compute and flexible continuation for higher compression ratios. We will clarify this better if the paper is accepted.
>
> Below we specify resource estimates for retrofitting the models used in the manuscript at 100 steps/CR to CR 8x. Please see the response to Q5 for details.
>
> | Model         | GPUh  |
> |---------------|------:|
> | Llama 3.2 1B  |    75 |
> | Llama 3.1 8B  |   700 |
> | R1-Qwen 1.5B  |   220 |
> | R1-Qwen 7B    |   530 |
> | R1-Qwen 32B   |  2400 |
>
> **W3: Practical evaluation on efficiency**
>
> **We provide latency measurements during generation using the Megatron-LM framework**. We run our measurements on H100 GPUs in bfloat16 precision using FlashAttention2 and the flash_attn_with_kvcache routine for both vanilla and DMS. We note that we use research code for DMS that has not been optimised (e.g., through fusion of multiple operations), resulting in a visible latency overhead. Also, for small to moderate token batch sizes, KV cache size does not drive the majority of latency (see Appendix H and Figure 6 for details). However, a practical inference scaling scenario of serving multiple users at a time and multiple rollouts per query (parallel scaling) requires substantial batch sizes. In addition, Qwen 2.5 models have smaller KV cache sizes than the Llama 3 family or the newer Qwen 3 family, when measured in relation to the number of model parameters. Finally, we remark that our Megatron-LM framework setup is designed for fast inference, albeit we could not optimise it to the point of matching state-of-the-art inference engines due to time constraints of the response period.
>
> For latency benchmarks, we measure the generation of reasoning chains for AIME24. The notation “Lat @ X” denotes the latency when generating a token at position X. From the results below, it emerges that even for moderate batch sizes (such as 64 and 128) and sequence lengths (>-8k), CR4 and CR8 decrease latency considerably. The reason why there is not a direct correspondence between the reduction in memory reads (due to higher CR) and reductions in latency is that their dependency emerges only at large token batch and sequence sizes. We refer to Appendix H for an explanation with heatmaps. Overall, **these findings suggest that DMS results in effective latency reductions due to decreased KV cache traffic in regimes such as inference-time scaling (parallel or sequential)**. In turn, this enhances reasoning quality by allowing for a larger effective token budget for the same wall-clock runtime (while also reducing memory load).
>
> | Model | Batch size | Lat @ 4k | Lat @ 8k | Lat @ 12k | Lat @ 16k |
> |-|-|-:|-:|-:|-:|
> | R1-Qwen 1.5B Vanilla | 16 | 19.1 ms | 19.3 m | 18.9 ms | 18.7 ms |
> | R1-Qwen 1.5B DMS 4x | 16 | 27.5 ms | 26.1 ms | 26.2 ms | 26.3 ms |
> | R1-Qwen 1.5B DMS 8x | 16 | 26.6 ms | 26.3 ms | 26.4 ms | 25.5 ms |
> | R1-Qwen 1.5B Vanilla | 64 | 21.9 ms | 39.0 ms | 40.7 ms | 41.1 ms |
> | R1-Qwen 1.5B DMS 4x | 64 | 26.2 ms | 26.3 ms | 25.9 ms | 26.4 ms |
> | R1-Qwen 1.5B DMS 8x | 64 | 25.5 ms | 25.2 ms | 24.9 ms | 24.9 ms |
> | R1-Qwen 1.5B Vanilla | 128 | 23.1 ms | 41.0 ms | 58.4 ms | 76.0 ms |
> | R1-Qwen 1.5B DMS 4x | 128 | 26.0 ms | 25.8 ms | 26.2 ms | 26.1 ms |
> | R1-Qwen 1.5B DMS 8x | 128 | 26.7 ms | 26.1 ms | 26.1 ms | 25.9 ms |
>
>
> | Model | Batch size | Lat @ 4k | Lat @ 8k | Lat @ 12k | Lat @ 16k |
> |-|-:|-:|-:|-:|-:|
> | R1-Qwen 7B Vanilla | 16 | 24.9 ms | 25.7 ms | 26.4 ms | 27.1 ms |
> | R1-Qwen 7B DMS 4× | 16 | 25.9 ms | 25.8 ms | 25.8 ms | 26.3 ms |
> | R1-Qwen 7B DMS 8× | 16 | 26.7 ms | 27.2 ms | 27.4 ms | 26.5 ms |
> | R1-Qwen 7B Vanilla | 32 | 25.8 ms | 42.8 ms | 44.4 ms | 45.0 ms |
> | R1-Qwen 7B DMS 4× | 32 | 25.7 ms | 25.9 ms | 25.9 ms | 25.6 ms |
> | R1-Qwen 7B DMS 8× | 32 | 26.0 ms | 26.9 ms | 26.9 ms | 27.1 ms |
> | R1-Qwen 7B Vanilla | 64 | 26.8 ms | 44.5 ms | 62.2 ms | 79.7 ms |
> | R1-Qwen 7B DMS 4× | 64 | 26.1 ms | 26.8 ms | 26.8 ms | 26.4 ms |
> | R1-Qwen 7B DMS 8× | 64 | 26.2 ms | 27.0 ms | 26.8 ms | 26.0 ms |
>
>
> We will include latency plots for all model sizes in the final version of our paper.
>
> **Q1: Extraction of the decision variable from $q[0]$**
>
> We agree with the Reviewer that the role of $q[0]$ in predicting the eviction decision should be better clarified in Section 3, we will adopt this suggestion in the next version of our paper. The reason why it was only introduced in the experimental setup is that this is not part of the “core” design of our method. In principle, alternative approaches exist, such as introducing a trainable vector $w$ and computing the decision variable as $x^Tw$. Reusing q[0] or k[0] is primarily a form of performance optimisation.
>
> **Q2: Ambiguous L, H, T symbols on L149**
>
> Thank you for bringing this to our attention. We will revise the formula using $|L|$, $|H|$, and $|T|$ to denote the number of layers, attention heads, and sequence length, respectively.
>
> **Q3: Avoiding materialisation of the attention mask**
>
> As shown in Figure 1(b), the attention mask can be reconstructed from a vector $\alpha = [log(1 - \alpha_1), \log(1 - \alpha_2), …, \log(1 - \alpha_n)]^T$ , as these values are replicated as columns of the mask, with the diagonal and sub-diagonals zeroed to account for the sliding window. This regular structure can be leveraged for a more efficient implementation of self-attention. A detailed discussion of a similarly compressed representation for a family of structured masks—which our mask belongs to—is given in [1], along with source code based on FlashAttention 2. We will convey this better in the text and reference this work in the final version of our manuscript.
>
> [1] G. Wang, J. Zeng, X. Xiao, S. Wu, J. Yang, L. Zheng, Z. Chen, J. Bian, D. Yu, H. Wang, FlashMask: Efficient and Rich Mask Extension of FlashAttention, 2025.
>
> **Q4: Only one q[0] is retrofitted for a GQA group, so are the eviction decisions consistent for all heads within a group?**
>
> Our models use GQA, where keys are shared among queries. Extracting the decision variable from a single query for each KV group impacts only one query-key pair, whereas extracting the variable from k[0] would impact all query-key pairs in which this key is used. We emphasise that all queries within the GQA group share the same KV cache after eviction.
>
> **Q5: Method not being robust in Figure 4 right for win=16**
>
> The win=16 result highlights that DMS can evict even very recent tokens effectively—contrary to prior assumptions—when delayed eviction is used. That said, a larger window (e.g., win=256) yields better data efficiency and makes the model operate more like vanilla, which results in overall better accuracy (as seen in Figure 4, right). Importantly, since CR is enforced over the entire context (e.g., 4k or 8k), increasing the window size should not increase memory use. We ran a series of post-submission experiments confirming that enlarging the window to 256 tokens does improve the results in reasoning models, and allows for reducing the number of retrofitting steps by an additional factor of 30x from 3000/CR to 100/CR. We will clarify this new practical insight and update the results in Figures 2 and 3. Below we show an excerpt from the results for R1-Qwen 7B in these two setups: (win=16, 3000 steps/CR) and (win=256, 100 steps/CR). AIME24 scores and KV budgets have been averaged over 3 runs with different random seeds due to the small dataset size.
>
> | Model | L-W-CR | AIME24 Score | AIME 24 KV Reads | MATH500 Score | Math 500 KV Reads | GPQA ◇ Score | GPQA ◇ KV Reads |
> |-:|-:|-:|-:|-:|-:|-:|-:|
> | Vanilla | 8-4-1 | 50.00 | 9.27E+07 | 93.20 | 2.21E+07 | 50.33 | 8.98E+07 |
> | DMS (win=256) | 8-4-4 | 60.00 | 3.02E+07 | 93.40 | 8.54E+06 | 54.00 | 3.16E+07 |
> | DMS (win‑16) | 8-4-4 | 46.67 | 2.63E+07 | 93.00 | 6.69E+06 | 53.87 | 2.65E+07 |
> | Vanilla | 32-1-1 | 53.33 | 8.84E+07 | 94.00 | 1.24E+07 | 51.50 | 4.54E+07 |
> | DMS (win=256) | 32-1-4 | 53.30 | 2.52E+07 | 92.80 | 3.66E+06 | 48.50 | 1.39E+07 |
> | DMS (win=16) | 32-1-4 | 44.43 | 2.07E+07 | 93.00 | 3.44E+06 | 49.50 | 1.22E+07 |
> | DMS (win=256) | 16-8-8 | 73.30 | 8.03E+07 | 96.60 | 1.62E+07 | 56.10 | 6.33E+07 |
> | DMS (win=16) | 16-8-8 | 62.23 | 6.87E+07 | 95.80 | 1.16E+07 | 52.17 | 5.10E+07 |
>
> The table shows that win=256, even with 30x shorter retrofitting, yields much better models.
>
> As for the results in Figure 4 right for win=16, Llama 3.2 1B Instruct model proved brittle during retrofitting, with GSM8K scores degrading quickly (e.g., DMC at CR 3x and 4x in Table 1). We use "robust" to mean the model still performs reasonably well and does not collapse the score under difficult settings (e.g., win=16 and low token budgets), but also maintains strong results with larger windows like win=256.
>
> **L1: Multiple runs per CR and data efficiency**
>
> Please see our response to W2 and Q5, which hopefully assuages the Reviewer’s concerns, since only a single run is needed for all CRs. In addition, we pushed data efficiency to 60x the previous SOTA trainable KV cache compression (DMC)
>
> **L2: Showing latency measurements gains instead of proxy metrics**
>
> We refer to our latency measurements reported in our response for W3. In a nutshell, DMS translates into effective latency reduction in regimes with large batch sizes and sequence lengths, which are common in inference-time scaling..

---

> > ### Comment · Reviewer_BS8X · 2025-08-06
> >
> > Thank you. All of my concerns have been addressed.

---

### Official Review · Reviewer_1K9q · 2025-07-03

**Clarity:** 3
**Significance:** 3
**Originality:** 3
**Rating:** 4
**Confidence:** 4

**Summary:**

This paper introduce Dynamic Memory Sparsification (DMS), a method for sparsifying KV cache by evicting tokens towards a target compression ratio. The core idea is that even if a token is decided to be evicted at some time t, the actual eviction is delayed for a later generation step t + w. Eviction decision probability is sampled from a Gumbel-sigmoid distribution as a function of constants and the first component of the query vector at time t. During inference this probability is binarized. During training it enters a summation over layers, attention heads and the sequence head so that it that the one-sided L1 loss from the target compression ration over them is minimized; also the model is retrofitted through logit distillation.

Experiments over AIME24, MATH500 and GPQA Diamond problems are then conducted for vanilla Deepseek-R1 distilled models of various sizes, their DMS-retrofitted models and for different training-free KV-cache sparsification methods (TOVA, H2O, Quest) under varying token budget configurations (in terms of maximum sequence length (L), number of parallel reasoning chains (W) and compression ratios (CR)). The accuracy as a function of runtime efficiency (KV cache token reads) and out-of-memory risk (peak tokens in memory) for different L-W-CR configurations can identify Pareto frontiers for each method. DMS achieves superior frontiers over benchmarked approaches with absolute gains of up to 10 points over vanilla methods, i.e. much improved accuracy of scaled inference as more tokens for the same compute budget can be generated thanks to the presented KV-cache compression scheme (inference-time hyper-scaling). Ablation studies over a small instruct model further elaborate the role of the delay window w (down to the immediate eviction scenario). Additionally DMS is applied on more tasks with particularly good performance on long-context ones.

**Questions:**

1. Can different retrofitted models (different context lengths and different compression rations) share checkpoints during their training process?

2. Can you identify scenaria where one of the training-free KV-cache compression methods would be more attractive? Are such considerations included in limitation sections?

**Ethical Concerns:**

["NO or VERY MINOR ethics concerns only"]

**Final Justification:**

Overall, I found the idea of "delaying token eviction" simple to present and integrate/implement and empirically effective; also the authors provided useful clarifications during rebuttal. Therefore, I maintain my initial positive score (4: borderline accept).

**Limitations:**

Yes

**Quality:**

3

**Strengths And Weaknesses:**

+ The idea is simple (delaying token eviction), yet particularly effective.

+ Integrating the core idea of delayed token eviction can be conveniently implemented with an additive attention mask.


- Models need to be retrofitted, which involves training.

- Training requirements seem to be a function of target context lengths and compression ratio (CR), so multiple model checkpoints need to be fine-tuned.

---

> ### Author Rebuttal · Authors · 2025-07-31
>
> We appreciate Reviewer 1K9q’s taking the time to comment on our paper and would like to thank them for their questions and remarks.
>
> **Q1: Can different retrofitted setups (context lengths and CRs) share checkpoints during retrofitting?**
>
> Yes indeed! A single DMS retrofitting run produces a sequence of checkpoints with increasing compression ratios. The target CR increases linearly from 1x to the preset maximum, and intermediate checkpoints (e.g., 2x, 3x) are saved along the way. The DMS models in Figures 2 and 3 were generated from single retrofitting runs per model size.
>
> The answer for context length has an additional nuance. For Figures 2 and 3, we trained R1-Qwen models with a fixed 8k context and evaluated them by generating sequences beyond this length, such as 16k and 32k. Hence, the DMS retrofitting procedure did not affect length extrapolation (possibly due to taking only a few steps and teaching to evict recent tokens).
>
> **Q2: In which scenarios training-free KV cache compression methods would be more attractive?**
>
> Apart from the main advantage, which is the lack of need for extra training, training-free methods might offer advantages in particular edge cases. Sliding-window methods like TOVA maintain full KV cache fidelity until the window is exceeded, preserving vanilla accuracy on short inputs. This stands in contrast with DMS, which compresses all sequences to some extent, albeit it compresses shorter sequences to a ratio inferior to the target (and longer ones to a ratio superior to the target). For the same reason, methods like TOVA allow for pre-specifying the memory cap, whereas the KV cache in DMS grows as a function of the sequence length.
>
> Nonetheless, for most intents and purposes, retrofitting is superior to training-free alternatives, as these do not offer a statistical guarantee of preserving performance across tasks and often lead to significant degradations [1]. We will include all these considerations in the Limitations section.
>
> [1] P. Nawrot, R. Li, R. Huang, S. Ruder, K. Marchisio, E.M. Ponti, The Sparse Frontier: Sparse Attention Trade-offs in Transformer LLMs, 2025.

---

> > ### Comment · Reviewer_1K9q · 2025-08-07
> >
> > Thank you for the clarifications. I will be maintaining my positive view of this work during further internal discussions towards finalization of evaluation and scores.

---

### Note · Authors · 2025-08-13

We thank all reviewers for their constructive feedback. Throughout the rebuttal and discussions, we have provided the following explanations and additional experiments to address their concerns:

 - **Cost of retrofitting.** Through additional experiments, we showed that training steps can be reduced even further by a factor of 30x provided a sufficiently large window (win=256). In addition, we clarified that _multiple compression ratios are obtained from a single training run._ Overall, this achieves unprecedented sample efficiency and alleviates concerns on the cost of retrofitting from Rs 1K9q BS8X and eA55, as shown by the breakdown of GPU hours by model.
 - **Latency gain measurements.** As requested by Rs BS8X and 2bQ1, we provided latency measurements in addition to proxy metrics such as memory reads, finding that _DMS significantly reduces latency during inference for large batch sizes (parallel scaling) and long sequences (sequential scaling)._ Crucially, vanilla and DMS models used the same, off-the-shelf FlashAttention2 routines optimized for inference.
 - **Broader task generalization:** While our primary focus was on long-sequence reasoning tasks, following R 2bQ1’s suggestion, we have extended our evaluation to include code generation (LiveCodeBench) and long-document question answering (QASPER). This new evidence supports the view that _DMS generalizes well, by maintaining performance on long-context tasks and enabling hyper-scaling on reasoning tasks._
 - **Writing clarity:** We are committed to improving notation and plot readability as suggested by R BS8X. In addition, we will elaborate on the scenarios where one of the training-free KV-cache compression methods would be more attractive (1K9q), the motivations behind different W-L-CR setups (eA55), and a dedicated limitations section (eA55). We will revise the manuscript accordingly.

We thank the reviewers for highlighting a series of strengths, including the novelty, simplicity, and effectiveness of DMS, a thorough and extensive evaluation, and substantive gains on the Pareto frontiers. **During the discussion, the reviewers also confirmed that all their concerns have been addressed or alleviated and that they retain a positive outlook on our paper.** We thank the reviewers and Area Chair once again for their efforts. The discussion has shaped our work into a more complete and sound contribution.

---

### Decision · Program_Chairs · 2025-09-17

**Decision:**

Accept (poster)

**Comment:**

(a) Summary of claims and findings:
The paper tackles the KV-cache bottleneck in inference-time scaling for LLMs and proposes Dynamic Memory Sparsification (DMS), a retrofittable KV compression method that delays token eviction by a sliding window before enforcing sparsity. The delayed eviction aims to preserve salient information while achieving a target compression ratio (CR), enabling “hyper-scaling”: generating more tokens or parallel chains under the same compute/memory budget to improve reasoning accuracy. Training uses a sparsity objective plus logit distillation; at inference a structured additive mask enforces eviction. Across several reasoning benchmarks (AIME24, MATH500, GPQA Diamond) and model families (DeepSeek R1-distilled/Qwen, Llama 3.2 1B), DMS improves accuracy–efficiency Pareto frontiers compared to training-free methods (TOVA, H2O, Quest) and prior trainable compression (DMC). During discussion, authors clarified that a single retrofitting run produces multiple CR checkpoints, introduced a larger window (win=256) that greatly reduces steps per CR (~30×) while improving stability, provided H100 latency measurements showing gains in large-batch/long-sequence regimes, and added broader task results (LiveCodeBench code generation, QASPER long-context QA).

(b) Strengths:

- Relevance and impact: KV cache is a primary limiter for inference-time scaling; improving the accuracy–efficiency frontier is practically valuable.
- Simple, implementable idea: delayed eviction via a structured mask that integrates with standard kernels; compatible with GQA.
- Strong empirical evidence on reasoning-heavy tasks and multiple model sizes; consistent Pareto improvements in KV reads/peak tokens versus accuracy.
- Practicality improvements substantiated in rebuttal: single-run, multi-CR retrofitting; win=256 dramatically improves sample efficiency and stability; latency results on H100 show real gains in realistic scaling regimes.
- Clear positioning relative to training-free and prior trainable approaches; ablations on window size and data budget provide actionable guidance.

(c) Limitations and remaining gaps:

- Novelty is evolutionary: DMS is a pragmatic variant of prior trainable KV compression (e.g., DMC), exchanging “merge” for “delayed evict.” Nonetheless, the delayed strategy is simple, effective, and easier to implement.
- Practical efficiency results rely on research code (non-optimized); latency wins are most visible at large batch/long sequence lengths. Production-engine validation (e.g., vLLM/TensorRT-LLM) would strengthen the case.
- Broader task coverage is still limited beyond reasoning; added results on code gen and long-context QA are helpful but brief.
Some methodological details (decision signal design, mask materialization, per-head policy) were clarified in rebuttal and should be surfaced in the main text; figure labeling (W–L–CR) can be tightened.

(d) Reasons for decision:

The paper addresses an important and timely bottleneck with a conceptually simple mechanism that consistently improves the accuracy–efficiency trade-off, enabling more effective inference-time scaling. The method is practical to integrate, the evaluation on reasoning tasks is thorough, and the rebuttal substantially strengthens concerns on retrofitting cost (single run, multi-CR; win=256 for ~30× fewer steps), latency (H100 measurements), and generalization (code, long-context QA). While the contribution is more systems/pragmatics than theoretical novelty, the demonstrated gains and practicality justify acceptance as a poster.

(e) Suggestions for camera-ready:

- Move key clarifications into the main text: single-run multi-CR procedure, win=256 data-efficiency/stability improvements, decision-variable options (q[0] vs k[0] vs learnable probe) with brief ablations, GQA implications, and mask implementation (e.g., relate to FlashMask).
- Add a concise end-to-end retrofit cost table (tokens, GPU-hours) per model/CR/window; include variance across seeds and practical heuristics for choosing window size and CR ramps.
- Summarize H100 latency results in the main paper and, if feasible, add a small study on a production inference stack (throughput/VRAM vs batch/length).
- Improve figure readability for the W–L–CR trade-off plots; add an explicit limitations section noting regimes where training-free methods may be preferable (short inputs, strict memory caps).
- Keep the new broader-task results (LiveCodeBench, QASPER) and briefly outline future stress tests (e.g., adversarial/irregular attention patterns).